# A FANO-STYLE ACCURACY UPPER BOUND FOR LLM SINGLE-PASS REASONING IN MULTI-HOP QA

**Kaiyang Wan[2,1], Lang Gao[1], Honglin Mu[1], Preslav Nakov[1], Yuxia Wang[2,1], Xiuying Chen[1]***

[1]MBZUAI, [2]INSAIT, Sofia University "St. Kliment Ohridski"
Xiuying.Chen@mbzuai.ac.ae

## ABSTRACT

Multi-Hop Question Answering (MHQA) requires integrating dispersed, interdependent evidence through sequential reasoning under noise. This task is challenging for LLMs as they have a finite per-pass output capacity, beyond which the integration of task-relevant evidence proves unreliable. Consequently, the single-pass reasoning paradigm is inherently vulnerable to this capacity overflow. To formalize this bottleneck, our analysis establishes a Fano-style accuracy upper bound, defining a theoretical performance ceiling for single-pass LLMs. This bound reveals that accuracy inevitably collapses once task complexity exceeds model capacity, providing general principles for capacity-aware representation and structuring of MHQA in LLMs. Building on these principles, we introduce a proof-of-concept multi-call framework for MHQA, InfoQA. It ensures high per-step accuracy by combining capacity-aware task decomposition with active pruning of prior reasoning traces, keeping the information load within the single-pass limit. It further achieves robustness by a dependency-explicit workflow that enables precise control over the reasoning path. We construct a stringent and noise-rich benchmark to validate our theory and framework. Experimental results show that model behavior aligns with our predicted capacity curves while InfoQA achieves consistent performance improvements. We hope our work inspires more LLM multi-step reasoning methods: ⏻InfoQA.

## 1 INTRODUCTION

Multi-Hop Question Answering (MHQA) (Yang et al., 2018; Trivedi et al., 2022; Mavi et al., 2024) is an important NLP task with critical applications in real-world domains such as scientific literature analysis and complex fact verification (Yin et al., 2023; Yu et al., 2021). The task requires integrating multiple, interdependent pieces of evidence that appear in different segments of a long provided context. As a result, solving MHQA demands compositional reasoning: the model must carry forward intermediate findings from one evidence source and use them to locate or interpret information in subsequent sources. This stepwise dependency structure forms a reasoning chain, where the accuracy of each intermediate inference directly determines the correctness of the final answer. Accordingly, task success hinges on accurately resolving each reasoning hop while maintaining a coherent chain that faithfully composes intermediate findings into the final conclusion.

MHQA remains challenging for Large Language Models (LLMs) (Achiam et al., 2023; Bai et al., 2023; Liu et al., 2024) despite recent advances in prompting strategies and reasoning techniques (Havrilla et al., 2024). As shown in Figure 1(a), intuitively, because an LLM generates only a finite number of tokens in a single pass and each token has limited representational capacity, the model is constrained by an upper bound on the total information it can carry forward. This output capacity bound limits the amount of dispersed evidence that the model can reliably integrate at once. When the reasoning chain spans multiple evidence sources or when the context contains substantial irrelevant content, the total information load often exceeds this bound. As a result, the model becomes prone to capacity overflow, where relevant signals are diluted or overshadowed by noise, leading to inaccurate intermediate inferences and, consequently, incorrect final answers.

---

*Corresponding author.

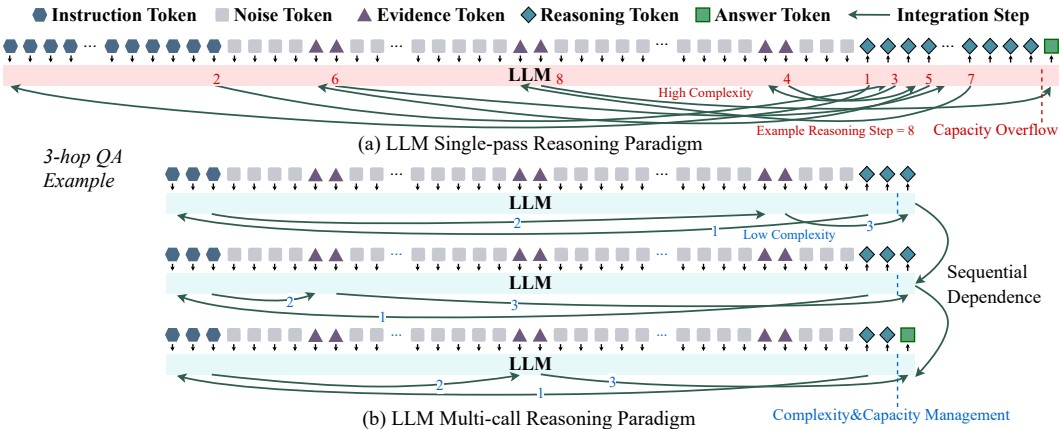

Figure 1: Comparison of single-pass and multi-call reasoning paradigms. Single-pass reasoning is constrained by the limited output capacity of LLMs, making it difficult to solve long-context and multi-hop problems. Multi-call reasoning mitigates this by decomposing tasks into sequentially dependent sub-steps, ensuring high per-step accuracy and a reliable reasoning chain.

To formalize this intuition, we first present an information-theoretic analysis that derives a Fano-style accuracy upper bound for LLM single-pass reasoning. This analysis reveals the *Accuracy Cliff*: when the task's information demand surpasses the model's output capacity, performance does not degrade gracefully but instead collapses sharply. We then examine why MHQA tasks are particularly prone to exceeding this cliff. By formalizing and dissecting the task structure, we identify two compounding challenges: Stepwise Capacity Overflow, driven by the super-linear growth of information demand with hop count and context length, and Cross-Step Error Accumulation, stemming from the amplification of even small per-step errors along the reasoning chain. Together, these analyses demonstrate that the single-pass paradigm is fundamentally inadequate for MHQA, motivating the design of a capacity-aware, multi-call paradigm as shown in Figure 1(b).

Building on the identified single-pass limitations and the structural demands of MHQA, we introduce InfoQA, a proof-of-concept multi-call framework for MHQA. InfoQA serves to concretely demonstrate how multi-call reasoning alleviates the dual crises of Stepwise Capacity Overflow and Cross-Step Error Accumulation. It does so by *(i)* capacity-aware task decomposition, which lowers the information demand and secures per-step accuracy, *(ii)* a dependency-explicit workflow, which enforces alignment across reasoning steps and prevents the chain from drifting off course, and *(iii)* iterative query contraction, which condenses the problem state and filters noise to keep information load manageable.

To precisely control hop count and context length, and thereby modulate the task-side information demand, we construct a dedicated dataset to test our theory. Experiments confirm that single-pass methods indeed exhibit an *Accuracy Cliff*, with results closely matching our theoretical curves. Moreover, as a proof-of-concept, InfoQA consistently outperforms single-pass baselines, further demonstrating the practical advantage of multi-call reasoning.

Our contributions can be summarized as follows:

1. We provide a rigorous information-theoretic analysis of LLM single-pass reasoning, deriving a Fano-style accuracy upper bound and revealing the *Accuracy Cliff* phenomenon (Section 2).

2. We dissect the structure of MHQA to explain why it is particularly prone to exceeding this limit, identifying two compounding challenges: Stepwise Capacity Overflow and Cross-Step Error Accumulation (Section 3).

3. We introduce InfoQA as a proof-of-concept in Section 4, and, in Section 5, we construct a controlled benchmark to validate our theory while demonstrating the practical advantage of multi-call reasoning paradigm.

## 2 THE INFORMATION BOTTLENECK IN LLM SINGLE-PASS REASONING

To analyze the inherent limits of single-pass LLM in complex reasoning, this section establishes a theoretical framework. We begin by formalizing the task and our analytical tools, then derive a universal accuracy upper bound that reveals a fundamental relationship between task complexity and model capacity.

### 2.1 FORMALIZING MHQA AND ANALYTICAL BASIS

**Problem Formulation.** We study MHQA in a *closed-book* setting, where the model must answer solely from the provided context. Formally, the input consists of a User Query $Q$ and a Context $C = (E, N)$, where $E = \{e_1, \ldots, e_M\}$ are the necessary evidence snippets and $N$ is irrelevant noise. The model generates an output $Y$, which includes its intermediate reasoning trace $R$ and the final answer tokens. An extractor $g$ then maps this output to the predicted answer $\hat{A} = g(Y)$.

**Analytical Basis.** Our analysis rests upon two foundational principles from information theory. We use $H(\cdot)$ to denote Shannon entropy (Shannon, 1948) and $I(\cdot; \cdot)$ for mutual information.

*1. Conditional Fano Inequality* (Fano & Hawkins, 1961). This principle establishes that to achieve a low error rate, the model's output must sufficiently resolve the initial uncertainty about the answer. It connects the error probability, $P_e = \Pr(\hat{A} \neq A \mid Q, C)$, to the residual uncertainty $H(A \mid Q, C, Y)$:

$$H(A \mid Q, C, Y) \ \leq \ h(P_e) + P_e \log(|\mathcal{A}| - 1). \tag{1}$$

*2. Output Entropy Bound* (Cover, 1999). This principle states that the amount of information an output $Y$ can provide about the answer $A$ is fundamentally capped by its own entropy. Formally, the mutual information is bounded as:

$$I(A; Y \mid Q, C) \ \leq \ H(Y). \tag{2}$$

We provide a more detailed discussion in Appendix A.2.

### 2.2 A FANO-STYLE ACCURACY UPPER BOUND

The performance of LLMs in single-pass reasoning is governed by a fundamental principle: the *information bottleneck*. Any single-pass output has a finite information-carrying capacity. When a task's complexity exceeds this capacity, a theoretical *performance ceiling* emerges, making ideal accuracy unattainable. By combining the Fano inequality with the output entropy bound from Section 2.1, we derive our central theorem, which forms the cornerstone of our framework.

**Theorem 1** (A Fano-Style Accuracy Upper Bound for Single-Pass Reasoning). *For any single-pass, closed-book policy, let $A \in \mathcal{A}$ be the ground-truth answer. Define the task's **information demand** as $\beta \triangleq H(A \mid Q, C)$ and the model's **output capacity** as $C \triangleq H(Y)$. The maximum achievable accuracy, $Acc = 1 - P_e$, is implicitly bounded by the following relationship:*

$$h(Acc) + (1 - Acc)\log(|\mathcal{A}| - 1) \ \geq \ \beta - C, \tag{3}$$

*where $h(\cdot)$ denotes the binary entropy function and $h(Acc) = h(1 - P_e)$.*

This theorem dictates that whenever the information demand $\beta$ of a task exceeds the output capacity $C$ of a model, achieving perfect accuracy ($Acc = 1$) becomes mathematically impossible.

### 2.3 FROM THEORY TO INTUITION: COROLLARIES AND THE ACCURACY CLIFF

While the exact bound in Theorem 1 is precise, its implications are more transparent through simplified corollaries. Together, they reveal a phenomenon we term the **Accuracy Cliff**.

**Linear Accuracy Bound.** By applying simple relaxations to the main theorem, we obtain a practical linear upper bound on accuracy:

$$Acc \ \leq \ \min\left\{1, 1 - \frac{\beta - C - 1}{\log |\mathcal{A}|}\right\}. \tag{4}$$

**Uniform-Distribution Case.** In the common scenario where the context makes all potential answers nearly equiprobable, the information demand simplifies to $\beta \approx \log|\mathcal{A}|$, and the general bound from Theorem 1 yields a more elegant and insightful upper bound on accuracy (proof in Appendix A.3):

$$Acc \;\leq\; \min\left\{1, \frac{C+1}{\beta}\right\}. \tag{5}$$

**Phase Transition and the Cliff Edge.** As shown in Figure 2 (taking $C = 200$ as an example), equation 5 describes the Accuracy Cliff curve . It reveals a sharp, phase-transition-like behavior: *(a) Capacity-Sufficient Regime ($\beta \leq C + 1$):* Before the critical threshold, the accuracy is capped at 1, where performance is perfect and stable. *(b) Capacity-Overflow Regime ($\beta > C+1$):* Immediately after this point, the performance ceiling collapses. The maximum achievable accuracy is no longer 1, but begins to decay hyperbolically according to the ratio $(C + 1)/\beta$. This transition from perfect accuracy to a rapid decay is the essence of the "Accuracy Cliff," illustrating how performance does not degrade gracefully but instead falls off sharply when the task complexity overwhelms the model's capacity.

Figure 2: The Accuracy Cliff. The theoretical upper bound on accuracy is plotted against information demand $\beta$, using $C = 200$ as an illustrative example. Once $\beta > C + 1$, the accuracy declines sharply.

This section establishes a universal performance bound that formalizes the fundamental limits of the single-pass reasoning paradigm. It proves that single-pass accuracy is ultimately constrained by an insurmountable barrier: the ratio of the task's information demand $\beta$ to the model's output capacity $C$. This insight does more than just explain existing failures; it shows the path forward. If single-pass reasoning is inherently bounded, the only viable solution is to transcend it. This theoretical bottleneck leads to the next critical questions: ***In a real-world MHQA setting, what factors cause the information demand $\beta$ to grow explosively? And how can we represent and structure the task to circumvent this single-pass limit?***

## 3 ANATOMY OF THE MULTI-HOP CHALLENGE

In this section, we provide a detailed dissection of the MHQA task, building on the Accuracy Cliff phenomenon from Section 2, to uncover the root causes of capacity overflow. The essence of MHQA is the navigation of a *latent reasoning chain*, represented as:

$$Z_0 \xrightarrow{\phi_1} Z_1 \xrightarrow{\phi_2} \cdots \xrightarrow{\phi_K} Z_K \xrightarrow{\phi_{K+1}} A.$$

In this chain, $Z_0$ is the initial entity from the query, $A$ is the final answer, and each intermediate $Z_k$ is a crucial "bridge" entity. The transformation $\phi_k$ represents the reasoning process itself that uses the context $C$ to advance from one entity to the next. This inherent chain structure is the source of a dual challenge: the risk of **Stepwise Capacity Overflow** within each individual step, and the systemic threat of **Cross-Step Error Accumulation** along the entire chain.

### 3.1 CHALLENGE 1: STEPWISE CAPACITY OVERFLOW

To predict when a model will be pushed off the Accuracy Cliff ($\beta > C$) established in Section 2, we now model the information demand $\beta$ as a function of task properties in MHQA.

**Modeling Task-Side Demand.** To connect our theoretical bound with observable task properties, we model $\beta$ as a function of hop count ($h$) and effective context length ($L$). Our model is based on three assumptions: (i) a *baseline complexity* $\beta_0$, representing the irreducible overhead of parsing a query and locating evidence in any single step; (ii) a *context burden* that scales linearly with context length ($L$) to reflect the worsening signal-to-noise ratio; and (iii) a *hop amplification* factor $\gamma^{h-1}$ ($\gamma \geq 1$) that captures the super-linear growth in complexity as uncertainty from prior steps propagates to subsequent ones. Combining these gives us the parametric form:

$$\beta(h, L) \;=\; \beta_0 \;+\; \alpha L \gamma^{h-1}. \tag{6}$$

This model shows that for $\gamma > 1$, $\beta$ grows super-linearly with the number of reasoning hops. This exponential growth is the primary driver that pushes a model toward the "Accuracy Cliff."

**Plug-in Accuracy Bound.** By substituting this demand model into equation 5, we get a concrete, testable prediction for how accuracy is limited by task characteristics:

$$Acc(h, L) \ \leq \ \min\left\{1, \frac{C+1}{\beta_0 + \alpha\, L\, \gamma^{\,h-1}}\right\}. \tag{7}$$

This equation formalizes a Capacity Crisis: as the number of hops $h$ or context length $L$ increases, the information demand $\beta$ escalates rapidly, heightening the likelihood of a capacity overflow $\beta > C$ and a consequent collapse in accuracy.

## 3.2 CHALLENGE 2: CROSS-STEP ERROR ACCUMULATION

The second challenge, Cross-Step Error Accumulation, arises not from the informational depth of any single step, but from the sequential nature of the reasoning chain itself. Even if the per-step accuracy is high, the overall probability of success can still collapse due to the amplification of small, individual errors as they propagate through the chain. To formalize this phenomenon, we first define a *stepwise success event*, $S_k$, where the model's prediction $\hat{Z}_k$ must be both correct and consistent with the prior state:

$$S_k \triangleq \big\{\hat{Z}_k = Z_k \ \wedge \ \hat{Z}_k = \phi_k(\hat{Z}_{k-1}, Q, C)\big\}, \quad (k = 1, \dots, K),$$

$$S_{K+1} \triangleq \big\{\hat{A} = A \ \wedge \ \hat{A} = \phi_{K+1}(\hat{Z}_K, Q, C)\big\}.$$

Overall success, $\mathsf{Succ} \triangleq \bigcap_{k=1}^{K+1} S_k$, therefore requires every step in the chain to succeed.

By the chain rule, $\Pr(\mathsf{Succ})$ is the product of the conditional success probabilities $p_k$ at each step:

$$\Pr(\mathsf{Succ}) = \prod_{k=1}^{K+1} \Pr\big(S_k \mid S_{<k}\big) = \prod_{k=1}^{K+1} p_k, \quad (8)$$

$$p_k = \Pr\Big(\hat{Z}_k = Z_k \ \wedge \ \hat{Z}_k = \phi_k(\hat{Z}_{k-1}, Q, C)\,\Big|\, S_{<k}\Big). \tag{9}$$

If we assume a uniform per-step success rate of at least $1 - \varepsilon$, the overall success probability is bounded by:

$$\Pr(\mathsf{Succ}) \ \geq \ (1 - \varepsilon)^{K+1} \ \approx \ 1 - (K+1)\varepsilon, \quad (10)$$

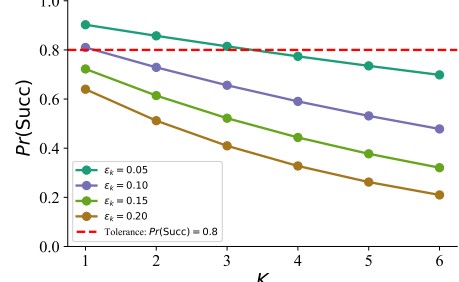

Figure 3: Error Accumulation. Even a small per-step error rate ($\varepsilon$) causes a rapid decay in overall success probability as the number of hops ($K$) increases.

This linear decay, visualized in Figure 3, formalizes the *Compounding Crisis*. It shows how the chain structure acts as an error amplifier. While the Capacity Crisis is the "spark" that generates individual errors, Cross-Step Error Accumulation is the "powder keg" that makes even small sparks catastrophic, causing the entire reasoning process to fail.

**An Inescapable Dilemma.** Built upon the above two challenges, our deconstruction of the multi-hop challenge reveals a dual, interlocking crisis rooted in its latent chain structure. The single-pass reasoning paradigm is thus caught in a vise grip: it is simultaneously vulnerable to *Stepwise Capacity Overflow*, which generates inevitable per-step errors, and to *Cross-Step Error Accumulation*, which guarantees that these errors will be catastrophically amplified. This dual-front assault renders the conventional single-pass paradigm fundamentally untenable for complex reasoning. Therefore, **the core issue is the very single-pass paradigm we force it into.**

## 4 INFOQA: A MULTI-CALL REASONING PARADIGM FOR MHQA

Our theoretical analysis in Section 2 established a universal performance limit for single-pass reasoning: the *Accuracy Cliff*, which dictates that accuracy inevitably collapses when information demand ($\beta$) exceeds model capacity ($C$).

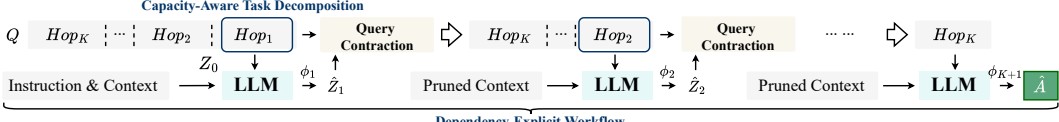

Figure 4: The InfoQA framework integrates three key components: (1) *Capacity-Aware Task Decomposition*, which reduces the information demand by generating single-hop sub-questions; (2) *Dependency-Explicit Workflow*, where the evolving contracted query carries the reasoning state across steps; and (3) *Iterative Query Contraction*, which prunes reasoning traces and rewrites the query with $\hat{Z}_k$. Each LLM call approximates $\phi_k$ and produces $\hat{Z}_k$.

Subsequently, our deconstruction of the MHQA task in Section 3 revealed exactly why this limit is so perilous in practice. We found that MHQA's structure not only causes $\beta$ to *escalate exponentially*, making capacity overflow almost certain, but also *catastrophically amplifies* the resulting errors along its reasoning chain. This dual diagnosis dictates the principles for an effective solution: a successful methodology must be both *capacity-aware* to manage per-step information load, and *robust* to maintain the integrity of the chain.

## 4.1 THE INFOQA FRAMEWORK

InfoQA is a multi-call reasoning framework designed from the ground up to navigate the dual crises of multi-hop reasoning. It operationalizes the principle of decomposition by breaking down a single, high-demand query into a sequence of capacity-aligned sub-tasks, each with a manageable information load. This is achieved through three synergistic components, as depicted in Figure 4.

**Capacity-Aware Task Decomposition.** The first step in InfoQA is to transform a high-level multi-hop question into a simpler, single-hop sub-question. This decomposition is critical for reducing the initial information demand $\beta = H(A \mid Q, C)$ to a more manageable per-step demand, $\beta_1 = H(Z_1 \mid Q, C)$. For a question such as: *"What is the birth date of the lead actor in the movie directed by the person who wrote 'Dune'?"*, the initial sub-question is generated as: *"Based on the provided context, who wrote 'Dune'?"* By focusing the LLM on this narrow task, we ensure the reasoning step remains well within its single-pass capacity $C$, thereby directly counteracting the *Capacity Crisis*.

**Dependency-Explicit Workflow.** Once the problem is decomposed, a critical challenge is to reliably link sequential steps, countering the *Compounding Crisis* described in equation 10. InfoQA achieves this with a *Dependency-Explicit Workflow*. Instead of relying on a model's internal memory, the workflow's state is explicitly maintained and passed as the *current, contracted query itself*. After finding $\hat{Z}_k$, the query $Q_k$ is updated to $Q_{k+1}$ by embedding this finding. For example: $Q_k$: "..., directed by the person who wrote 'Dune'?" $\rightarrow$ Finding: "Frank Herbert" $\rightarrow$ $Q_{k+1}$: "..., directed by Frank Herbert?". This makes the reasoning chain transparent, controllable, and robust against error propagation.

**Iterative Query Contraction.** This mechanism is the engine that ensures the information load remains low throughout the entire reasoning process. After each step, InfoQA contracts the problem state via two actions: *Pruning*, where the extensive reasoning trace is discarded to prevent noise accumulation, and *Contraction*, where the query is rewritten with the latest finding $\hat{Z}_k$. By iteratively pruning thoughts and contracting the query, we ensure the prompt for every step represents the most concise form of the *remaining* problem. This prevents prompt length from growing with reasoning depth, acting as the crucial enabler that protects the entire chain from *Stepwise Capacity Overflow*.

## 5 EXPERIMENTS

We conducted experiments to validate the two central claims of this work. Our evaluation is twofold: **1. Theory Validation:** We first tested whether the empirical performance of LLMs aligns with our theoretical *Fano-style accuracy upper bound*, confirming that the Accuracy Cliff is a real and predictable phenomenon. **2. Framework Validation:** We then evaluated whether InfoQA framework can effectively transcend this theoretical limit, alleviating the capacity bottleneck to yield substantial performance gains.

## 5.1 Experimental Setup

**Benchmark Construction.** Existing MHQA benchmarks are unsuitable for our study as they lack fine-grained control over task difficulty and are often compromised by data artifacts, preventing a rigorous test of our theory. We therefore constructed a new, stringent, and noise-rich synthetic benchmark guided by three core principles: *(i) systematic control* over information demand ($\beta$) by varying hop count and distractor scale; *(ii) high semantic similarity* between evidence and distractors to prevent shortcut learning; and *(iii) a path maximization strategy* for evidence placement to enforce genuine, non-trivial reasoning chains. This process yielded a suite of datasets with systematically varied hop counts and context lengths, allowing for a precise evaluation of model performance against our theoretical bounds. We provide the key statistics of our benchmark in Table 1 and detailed construction consideration and algorithm in Appendix A.4.

**Models and Baselines.** We conducted our experiments on the Qwen3-8B and -14B (Yang et al., 2025). We chose this publicly available model family to minimize architectural and training biases, allowing for a fair evaluation of the reasoning *paradigms* themselves. All re-

Table 1: Statistics of our synthetic multi-hop QA benchmark.

|  | 1-hop | 2-hop | 3-hop | 4-hop |
|---|---|---|---|---|
| **Context Length** $L$ | | [0.5k, 1k, 2k, 4k, 8k, 10k] | | |
| **Samples per** $L$ | 300 | 300 | 300 | 300 |
| **Total Samples** | 1,800 | 1,800 | 1,800 | 1,800 |
| **Evidence Order** | $[e_1]$ | $[e_2, e_1]$ | $[e_2, e_3, e_1]$ | $[e_2, e_4, e_3, e_1]$ |
| **Evidence Position** | $[1/2]$ | $[1/3, 2/3]$ | $[1/4, 2/4, 3/4]$ | $[1/5, 2/5, 3/5, 4/5]$ |
| **Grand Total** | | 7,200 | | |

sults were obtained via official API calls. For all methods, we set temperature to 0.2 and a maximum generation length of 4096 tokens. Other parameters were default. We compared InfoQA against a comprehensive suite of strong single-pass baselines, including: (i) Direct Prompting, (ii) Chain-of-Thought (CoT) (Wei et al., 2022), (iii) Self-Consistency (SC)[1] (Wang et al., 2023b), (iv) Self-Refine[2] (Madaan et al., 2023), (v) ReAct (Yao et al., 2023), (vi) Plan-and-Solve (Wang et al., 2023a), and (vii) Self-Ask (Press et al., 2023). All baseline prompts were implemented as zero-shot, single-pass methods, carefully designed to follow the principles laid out in their respective original papers. All LLM calls within the InfoQA framework used the same backbone model and inference settings as the baselines. We used F1 as the evaluation metric.

## 5.2 Empirical Validation of the Accuracy Cliff

The results of Qwen3-14B and Qwen3-8B showed the same phenomenon; we analyze Qwen3-14B and present Qwen3-8B in Appendix A.7. Table 2 summarizes the average F1 scores across different context lengths and hop counts of Qwen3-14B. Our first experimental goal is to validate our core theoretical claim: the performance of single-pass models in MHQA is governed by an *accuracy cliff*. Concretely, we tested whether the empirical performance of strong prompting baselines conforms to the Fano-style accuracy upper bound derived in Section 2.

**Parameter Estimation Protocol.** To connect theory with data, we fit the parameters $\theta = (\beta_0, \alpha, \gamma, C)$ of our plug-in accuracy bound (Eq. 7) to empirical F1 scores, using F1 as a proxy for accuracy, $\widehat{\mathrm{Acc}}(h, L) = \mathrm{F1}(h, L)$. We minimized the mean absolute deviation between the observations and the bound:

$$\min_{\theta} \sum_{(h,L)} \left| \widehat{\mathrm{Acc}}(h, L) - \min\left\{1, \frac{C+1}{\beta_0 + \alpha L \gamma^{h-1}}\right\} \right|. \tag{11}$$

For each baseline we conducted a fine-grained grid search over $(\alpha, \gamma, \beta_0, C)$ and select the minimizer with respect to MAE. The fitted curves were then overlaid with empirical points (F1) as a function of the fitted effective demand $\beta(h, L)$. We present the fitted plots in Figure 5, with detailed fitting statistics in Appendix A.6 and fitting algorithm in Appendix A.5.

**Alignment with Predicted Curves.** Three consistent patterns emerged. *(i) Accuracy cliff:* as the effective demand $\beta$ grows with hop count and context length, empirical points adhere closely to the theoretical bound and then collapse once $\beta \gtrsim C+1$, consistent with the predicted cliff.

---

[1] Our implementation of Self-Consistency involves generating five reasoning paths by querying the model with varying temperatures: {0.1, 0.3, 0.5, 0.7, 0.9}. The final answer is determined by a majority vote.

[2] For Self-Refine, we report the final answer after one iteration of feedback and refinement.

Table 2: Average F1 scores of Qwen3-14B across different reasoning depths and context lengths. We compare InfoQA with single-pass baselines: Chain-of-Thought (CoT), Self-Refine (S-R), Self-Consistency (S-C), ReAct, Plan-and-Solve (P&S), Self-Ask (S-A), and InfoQA with ablation: w/o Capacity-Aware Task Decomposition (D.) and w/o Pruning Past Reasoning Trace (P.).

| Hops | Context Length | Average F1 Score | | | | | | | | w/o D. | w/o P. |
|---|---|---|---|---|---|---|---|---|---|---|---|
| | | Direct | CoT | S-R | S-C | ReAct | P&S | S-A | InfoQA | | |
| **1** | 0.5k | **1.00** | **1.00** | **1.00** | **1.00** | **1.00** | **1.00** | **1.00** | **1.00** | 0.87 | **1.00** |
| | 1k | **1.00** | **1.00** | **1.00** | **1.00** | 0.99 | **1.00** | **1.00** | **1.00** | 0.78 | **1.00** |
| | 2k | 0.99 | **1.00** | 0.99 | **1.00** | 0.99 | **1.00** | 0.99 | **1.00** | 0.79 | **1.00** |
| | 4k | 0.97 | 0.99 | 0.98 | **1.00** | 0.97 | 0.98 | 0.97 | 0.99 | 0.63 | **1.00** |
| | 8k | 0.93 | 0.98 | 0.82 | **1.00** | 0.72 | 0.89 | 0.93 | 0.98 | 0.31 | 0.96 |
| | 10k | 0.91 | 0.96 | 0.79 | 0.98 | 0.59 | 0.84 | 0.85 | 0.96 | 0.28 | 0.90 |
| **2** | 0.5k | 0.78 | **1.00** | **1.00** | **1.00** | **1.00** | **1.00** | 0.84 | **1.00** | 0.85 | **1.00** |
| | 1k | 0.74 | **1.00** | 0.98 | **1.00** | **1.00** | **1.00** | 0.75 | **1.00** | 0.84 | **1.00** |
| | 2k | 0.66 | **1.00** | 0.94 | **1.00** | 0.99 | 0.98 | 0.69 | **1.00** | 0.83 | **1.00** |
| | 4k | 0.54 | 0.99 | 0.77 | 0.99 | 0.96 | 0.85 | 0.68 | **1.00** | 0.84 | 0.98 |
| | 8k | 0.23 | 0.79 | 0.39 | 0.83 | 0.53 | 0.54 | 0.63 | **0.96** | 0.52 | 0.88 |
| | 10k | 0.18 | 0.76 | 0.44 | 0.81 | 0.55 | 0.60 | 0.63 | **0.89** | 0.39 | 0.83 |
| **3** | 0.5k | 0.70 | 0.97 | 0.95 | **0.98** | **0.98** | **0.98** | 0.85 | **0.98** | 0.93 | **0.98** |
| | 1k | 0.55 | 0.97 | 0.83 | **0.98** | 0.96 | 0.92 | 0.75 | **0.98** | 0.80 | 0.96 |
| | 2k | 0.41 | 0.94 | 0.66 | **0.97** | 0.84 | 0.83 | 0.74 | 0.96 | 0.67 | 0.94 |
| | 4k | 0.31 | 0.72 | 0.30 | 0.77 | 0.59 | 0.61 | 0.64 | **0.84** | 0.60 | 0.79 |
| | 8k | 0.06 | 0.32 | 0.12 | 0.35 | 0.24 | 0.19 | 0.52 | **0.64** | 0.43 | 0.44 |
| | 10k | 0.04 | 0.27 | 0.10 | 0.26 | 0.20 | 0.15 | 0.39 | **0.42** | 0.29 | 0.39 |
| **4** | 0.5k | 0.26 | 0.98 | 0.90 | **0.99** | 0.96 | 0.96 | 0.94 | 0.96 | 0.92 | 0.95 |
| | 1k | 0.13 | 0.95 | 0.79 | **0.98** | 0.87 | 0.93 | 0.84 | 0.96 | 0.84 | 0.92 |
| | 2k | 0.09 | 0.77 | 0.46 | 0.80 | 0.64 | 0.66 | 0.76 | **0.95** | 0.75 | 0.83 |
| | 4k | 0.02 | 0.49 | 0.34 | 0.54 | 0.41 | 0.38 | 0.55 | **0.93** | 0.56 | 0.69 |
| | 8k | 0.00 | 0.17 | 0.13 | 0.21 | 0.13 | 0.16 | 0.36 | **0.69** | 0.32 | 0.36 |
| | 10k | 0.00 | 0.09 | 0.09 | 0.12 | 0.06 | 0.06 | 0.21 | **0.30** | 0.23 | 0.18 |
| **Overall Average (2–4 hop)** | | 0.32 | 0.73 | 0.57 | 0.75 | 0.66 | 0.66 | 0.65 | **0.86** | 0.65 | 0.78 |
| | **1 hop Average** | 0.97 | 0.98 | 0.93 | **0.99** | 0.88 | 0.95 | 0.96 | **0.99** | 0.61 | 0.98 |
| | **2 hop Average** | 0.52 | 0.92 | 0.75 | 0.94 | 0.84 | 0.83 | 0.70 | **0.97** | 0.71 | 0.95 |
| | **3 hop Average** | 0.34 | 0.70 | 0.49 | 0.72 | 0.63 | 0.61 | 0.65 | **0.80** | 0.62 | 0.75 |
| | **4 hop Average** | 0.09 | 0.57 | 0.45 | 0.61 | 0.51 | 0.53 | 0.61 | **0.80** | 0.60 | 0.65 |
| Context Average (2–4 hop) | | | | | | | | | | | |
| | 0.5k | 0.58 | 0.98 | 0.95 | **0.99** | 0.98 | 0.98 | 0.88 | 0.98 | 0.90 | 0.98 |
| | 1k | 0.48 | 0.97 | 0.87 | **0.99** | 0.94 | 0.95 | 0.78 | 0.98 | 0.83 | 0.96 |
| | 2k | 0.38 | 0.90 | 0.69 | 0.92 | 0.83 | 0.83 | 0.73 | **0.96** | 0.75 | 0.92 |
| | 4k | 0.29 | 0.73 | 0.47 | 0.77 | 0.65 | 0.61 | 0.62 | **0.92** | 0.67 | 0.82 |
| | 8k | 0.10 | 0.43 | 0.21 | 0.46 | 0.30 | 0.30 | 0.50 | **0.76** | 0.42 | 0.56 |
| | 10k | 0.07 | 0.37 | 0.21 | 0.40 | 0.27 | 0.27 | 0.41 | **0.54** | 0.30 | 0.47 |

***(ii) Capacity and hop inflation:*** CoT substantially increases the effective single-pass capacity $C$ and reduces hop inflation $\gamma$ relative to Direct, thereby delaying the onset of the cliff; S-C exhibits a similar trend. ***(iii) Method-specific overheads:*** certain methods introduce additional demand. For example, S-A shows a large $\beta_0$ (higher base demand), which offsets the benefit of a larger $C$. Overall, the fitted overlays corroborate these findings: empirical markers align tightly with the theoretical envelope at low $\beta$ and diverge only when the bound becomes active.

## 5.3 PERFORMANCE OF INFOQA

**Overall performance.** As shown in Table 2, InfoQA achieves the best results across most settings, with an overall average of 0.86 on 2–4 hop tasks, substantially outperforming strong single-pass baselines such as S-C (0.75) and CoT (0.73). The key strength of InfoQA lies in its robustness along two axes. First, in terms of *depth robustness*, InfoQA sustains high accuracy even as the hop count increases, whereas single-pass baselines suffer sharp degradation beyond 2 hops due to compounded informational demand and error accumulation. Second, in terms of *length robustness*, InfoQA remains reliable under long contexts (8k–10k tokens), while methods like Direct and ReAct collapse to near-zero. This stability comes from explicitly pruning past traces and contracting queries, which prevents context inflation and keeps the effective demand $\beta$ within the model's per-pass capacity $C$.

**Ablation study.** We further examined the contribution of InfoQA's key design choices: *(i) w/o Decomposition (w/o D.)*, which executed the full reasoning chain in a single-pass without capacity control, and *(ii) w/o Pruning (w/o P.)*, which preserved all past reasoning traces without contraction.

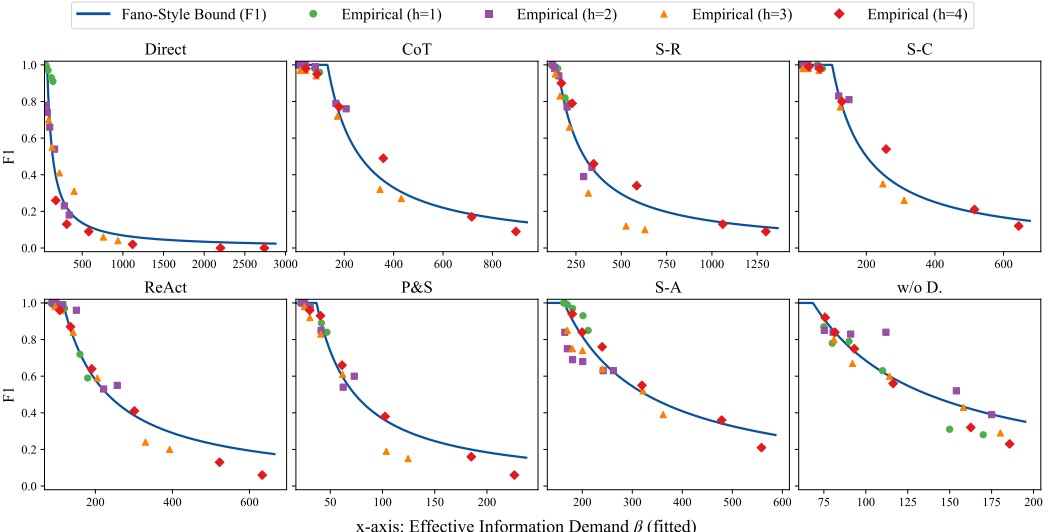

Figure 5: Qwen3-14B F1 vs. theoretical curves across single-pass methods. The x-axis shows the estimated effective information demand ($\beta$), fitted per method, and the y-axis shows the F1 score.

As shown in Table 2, w/o D. quickly saturated at longer contexts and higher hops (overall average 0.65), confirming the single-pass bottleneck predicted by the Accuracy Cliff. Meanwhile, w/o P. performed better but still trailed InfoQA (0.78 vs. 0.86), as unpruned traces inflated context length and exacerbated cross-step errors. These results highlighted that both *capacity-aware decomposition* and *iterative pruning* were indispensable: decomposition ensured per-step demand remained within capacity, while pruning prevented error amplification across the reasoning chain.

**Error Analysis of InfoQA.** Compared with single-pass baselines, InfoQA exhibits a distinct error profile. Since its multi-call design try to prevent capacity overflow, most residual failures are not caused by information bottlenecks but by *semantic drift* during iterative query contraction. In particular, the contracted query may sometimes omit subtle constraints (e.g., temporal qualifiers or entity disambiguation), causing the reasoning chain to pursue a plausible but incorrect path. Another source of failure lies in the *intrinsic model capacity*: even when the task is decomposed into single-hop sub-questions, extremely long contexts can exceed the model's base comprehension ability. Combined with multi-hop error accumulation, this results in degraded performance for InfoQA on long-context, high-hop scenarios. These errors suggest that future work should focus on better decomposition to minimize the sub-task demand, improving contraction fidelity, and improving model's base capacity.

## 6 RELATED WORK

**LLM Single-pass Prompting Methods.** Single-pass prompting methods ask the model to complete the entire reasoning process in one forward generation, without external decomposition or iterative calls. Classic examples (Kojima et al., 2022; Chen et al., 2025; Zamfirescu-Pereira et al., 2023) include Direct prompting, Chain-of-Thought (CoT) (Wei et al., 2022). More structured variants such as ReAct (Yao et al., 2023), Plan-and-Solve (Wang et al., 2023a), and Self-Ask (Press et al., 2023) guide the model with explicit prompting templates to elicit stepwise reasoning. Despite these design differences, all of them operate within a single forward pass, meaning that the reasoning chain must fit entirely within the model's per-pass information capacity. As a result, their performance inevitably degrades when task complexity exceeds this capacity. Our work formalizes and quantifies this single-pass capacity limit, showing that it gives rise to the "accuracy cliff" observed in MHQA task.

**Multi-call Methods.** In contrast to single-pass prompting, multi-call methods decompose reasoning into multiple model invocations, with each call addressing a sub-task. A representative line of work is Self-Refine (Madaan et al., 2023), which iteratively generates feedback and refines the answer.

Other approaches adopt recursive or pipeline-style reasoning, such as multi-step decomposition for question answering (Li et al., 2024), programming (Qian et al., 2024; Kim et al., 2024), fact checking (Xie et al., 2025) and writing (Shao et al., 2024; Wan et al., 2025). The success of these methods has empirically validated the effectiveness of distributing the reasoning load across multiple calls. Building on this paradigm, our work provides a theoretical foundation from an information capacity perspective to explain why such an approach is beneficial. We show that single-pass methods face an inherent capacity bottleneck and that multi-call reasoning can provably keep the per-step information demand below the model's capacity.

**Information-Theoretic Perspectives on MHQA.** Information theory is useful to analyze the challenges and bottlenecks of MHQA tasks. Xu et al. (2025) focused on retrieval-based systems, using pointwise conditional V-information to quantify the contribution of documents and optimize the retriever's selection process. Chen (2025) addressed the parameter storage capacity, establishing a theoretical lower bound on the number of parameters necessary to reliably store multi-hop reasoning chains within the model weights. Complementary to these retrieval and storage perspectives, our work targets the closed-book setting to formalize the single-pass output channel capacity bottleneck, identifying the Accuracy Cliff where performance collapses due to limited generation bandwidth rather than insufficient knowledge storage.

## 7    CONCLUSION AND FUTURE WORK

In this work, we began by providing an information-theoretic analysis of MHQA with LLMs. By deriving a Fano-style accuracy upper bound, we formalized the fundamental capacity bottleneck of single-pass reasoning and revealed the Accuracy Cliff, where accuracy collapses once information demand exceeds model capacity. Building on this insight, we dissected MHQA to identify the dual challenges of stepwise capacity overflow and cross-step error accumulation, showing why single-pass reasoning is inherently fragile. To validate our theoretical analysis, we introduced InfoQA, a capacity-aware multi-call proof-of-concept that decomposes complex queries into manageable steps, prunes noisy traces, and explicitly controls dependency flow. Our experiments results align with the predicted capacity curves and InfoQA achieves consistent gains.

Looking ahead, we believe this work opens several promising directions: First, extending our analysis to multi-call settings could clarify how information accumulates across calls and what new limits emerge. Second, adaptive decomposition strategies would let systems dynamically decide how to split queries based on complexity and improving model's base information capacity. Third, applying the capacity-bound perspective to domains such as science or law would test its robustness under real-world noise and reasoning demands.

## ETHICS STATEMENT

As part of our experimental design, we generated a synthetic dataset in which all personal names and company names are entirely fictitious. These synthetic entities do not correspond to real individuals or organizations. The use of fabricated identifiers was intentional, in order to avoid potential privacy, legal, or ethical concerns that could arise from using real-world data. No personally identifiable information (PII) or sensitive data were collected or used in this work. Therefore, we believe that our research does not pose risks to individuals, groups, or organizations.

## REPRODUCIBILITY STATEMENT

To ensure the reproducibility of our work, we provide an anonymous GitHub repository containing: (1) the synthetic dataset used in our experiments, (2) the code for constructing the dataset, (3) the implementation of all baselines as well as our proposed model, (4) the code used to fit empirical results to our theoretical curves, and (5) detailed README guidelines to facilitate reproduction of our results. All experiments can be reproduced directly using the provided resources. In addition, we have uploaded a compressed archive containing all these files as part of our paper submission, so that reviewers can access and reproduce our results even without relying on the external repository.

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

# A  APPENDIX

## A.1  LLM USAGE

We used LLMs as auxiliary tools during the preparation of this work. Specifically, LLMs were employed in three ways: (1) for proofreading and identifying minor typographical errors in the manuscript, (2) for generating a synthetic dataset that was used as part of our experiments, and (3) for automatic code completion during the development of our implementation. All research ideas, experimental design, and final manuscript writing remain the responsibility of the authors.

## A.2  INFORMATION-THEORETIC PRELIMINARIES: FULL PROOFS AND DISCUSSION

This appendix expands upon the information-theoretic preliminaries introduced in Section 2. We provide complete proofs of the conditional Fano inequality and the output entropy bound, together with intuitive interpretations and implications for multi-hop reasoning.

### A.2.1  PROOF OF THE CONDITIONAL FANO INEQUALITY

**Setup.** Let $A$ be the ground-truth answer, $\hat{A} = g(Y, Q, C)$ the prediction derived from the model output $Y$ (allowing the estimator to depend on $(Q, C)$), and $(Q, C)$ denote the query and context. Define the error event $E = \{\hat{A} \neq A\}$ with probability $P_e \triangleq \Pr(E = 1 \mid Q, C)$.

**Step 1: Decomposition of conditional entropy.** We begin from the chain rule of entropy:

$$H(A \mid Q, C, Y) = H(A, E \mid Q, C, Y) - H(E \mid A, Q, C, Y). \tag{12}$$

Since $E$ is a deterministic function of $(A, Y, Q, C)$, the last term vanishes, yielding

$$H(A \mid Q, C, Y) = H(E \mid Q, C, Y) + H(A \mid E, Q, C, Y). \tag{13}$$

**Step 2: Bounding each term.** By the fact that conditioning reduces entropy,

$$H(E \mid Q, C, Y) \leq H(E \mid Q, C) = h(P_e),$$

where $h(\cdot)$ is the binary entropy function. For the second term, conditioned on $E = 1$ (error), the uncertainty about $A$ is at most $\log(|\mathcal{A}| - 1)$, since all but the predicted answer remain possible. Thus

$$H(A \mid E, Q, C, Y) \leq P_e \log(|\mathcal{A}| - 1).$$

**Step 3: Combine.** Together, we obtain the bound:

$$H(A \mid Q, C, Y) \leq h(P_e) + P_e \log(|\mathcal{A}| - 1). \tag{14}$$

**Step 4: Mutual information form.** Rearranging yields the equivalent lower bound on mutual information:

$$I(A; Y \mid Q, C) \geq H(A \mid Q, C) - \big[h(P_e) + P_e \log(|\mathcal{A}| - 1)\big]. \tag{15}$$

$$\square$$

This bound states that unless the predictor extracts at least $\beta = H(A \mid Q, C)$ bits of information about $A$, a nontrivial error rate is unavoidable. In other words, *information demand implies error floor*.

### A.2.2 PROOF OF THE OUTPUT ENTROPY BOUND

**Setup.** The output $Y$ is a sequence of tokens from vocabulary $V$. We distinguish two modeling choices for the length constraint.

**Step 1: Mutual information bounded by entropy.** By definition and because conditioning reduces entropy,

$$I(A; Y \mid Q, C) \leq H(Y \mid Q, C) \leq H(Y).$$

**Step 2: Upper bounds on output entropy (two cases).**

- **Fixed length $m$ (or padded-to-$m$ with a special token).** Then $Y \in V^m$ and

$$H(Y) \leq \log |V|^m = m \log |V|. \tag{16}$$

- **Variable length, at most $m$ tokens (no padding).** Then $Y \in \bigcup_{k=0}^{m} V^k$ with cardinality $\sum_{k=0}^{m} |V|^k = \frac{|V|^{m+1} - 1}{|V| - 1}$, hence

$$H(Y) \leq \log\Big(\frac{|V|^{m+1} - 1}{|V| - 1}\Big). \tag{17}$$

Either equation 16 or equation 17 provides a valid capacity upper bound, depending on the modeling choice.

### A.2.3 IMPLICATIONS FOR MULTI-HOP REASONING

The two inequalities together establish an **information bottleneck** for single-pass reasoning:

- **Demand side ($\beta = H(A \mid Q, C)$).** Multi-hop QA inherently requires integrating dispersed and noisy evidence, which inflates the conditional entropy of the answer.
- **Supply side ($C = H(Y)$).** The single-pass output has a finite entropy budget, given by equation 16 or equation 17, scaling with output length and vocabulary.
- **Error floor ($P_e$).** Whenever $\beta > C$, Fano's inequality dictates that the error probability cannot vanish, regardless of model size or training.

This formalizes the intuitive statement: *"No matter how smart the model is, if the task demands more information than the output can encode, an error plateau is inevitable."*

### A.3 PROOF OF THE FANO-STYLE ACCURACY UPPER BOUND AND ITS COROLLARIES

**Notation and setup.** Throughout, all logarithms are base 2, so entropies and mutual information are measured in bits. We consider a *closed-book, single-pass* setting with a discrete answer space $\mathcal{A}$, $|\mathcal{A}| \geq 2$. The query $Q$ and context $C$ are given (conditioning variables). Let $A \in \mathcal{A}$ be the gold answer, $Y$ be the model's single-pass output (a random variable taking values in a finite or countable set of token sequences), and $\hat{A} = g(Y)$ be the predicted answer obtained by a *deterministic* extractor $g$. Define the error probability

$$P_e = \Pr(\hat{A} \neq A \mid Q, C), \text{ and } Acc = 1 - P_e.$$

We also define the task *information demand* $\beta \triangleq H(A \mid Q, C)$ and the model's *output capacity* $C \triangleq H(Y \mid Q, C)$. When needed, one may upper-bound $C$ by modeling constraints on $Y$: if $Y$ has fixed length $m$ (or is padded to $m$ with a special token) then

$$H(Y \mid Q, C) \leq m \log |V|;$$

if $Y$ has variable length at most $m$ without padding, then

$$H(Y \mid Q, C) \leq \log \left( \frac{|V|^{m+1} - 1}{|V| - 1} \right).$$

**Two ingredients.** We rely on two standard facts (made conditional on $(Q, C)$):

1. **Conditional Fano inequality** (e.g. Fano & Hawkins 1961, conditionalized on $(Q, C)$). For any estimator $\hat{A}$ of $A$,

$$H(A \mid Q, C, \hat{A}) \leq h(P_e) + P_e \log \left( |\mathcal{A}| - 1 \right), \tag{18}$$

where $h(\cdot)$ is the binary entropy function.

2. **Output-entropy (capacity) bound** (e.g. Cover 1999): for any $(A, Y)$,

$$I(A; Y \mid Q, C) \leq H(Y \mid Q, C) = C. \tag{19}$$

**A useful comparison between $Y$ and $\hat{A} = g(Y)$.** Because $\hat{A}$ is a deterministic function of $Y$, conditioning on the *richer* variable $Y$ cannot increase uncertainty relative to conditioning on $\hat{A}$:

$$H(A \mid Q, C, Y) \leq H(A \mid Q, C, \hat{A}). \tag{20}$$

Combining equation 20 with equation 18 yields

$$H(A \mid Q, C, Y) \leq h(P_e) + P_e \log \left( |\mathcal{A}| - 1 \right). \tag{21}$$

**Proof of Theorem 1.** Start from the chain rule for conditional mutual information:

$$I(A; Y \mid Q, C) = H(A \mid Q, C) - H(A \mid Q, C, Y) = \beta - H(A \mid Q, C, Y).$$

Apply equation 21 to upper-bound the second term:

$$I(A; Y \mid Q, C) \geq \beta - \left[ h(P_e) + P_e \log(|\mathcal{A}| - 1) \right].$$

Together with the capacity bound equation 19, we obtain

$$\beta - \left[ h(P_e) + P_e \log(|\mathcal{A}| - 1) \right] \leq I(A; Y \mid Q, C) \leq C.$$

Rearranging gives

$$h(P_e) + P_e \log(|\mathcal{A}| - 1) \geq \beta - C.$$

Finally, substitute $P_e = 1 - Acc$ and note that $h(P_e) = h(1 - Acc) = h(Acc)$ to obtain

$$h(Acc) + (1 - Acc) \log(|\mathcal{A}| - 1) \geq \beta - C, \tag{22}$$

which is Theorem 1. $\square$

**Derivation of the Linear Accuracy Bound (Eq. 4).** Starting from Theorem 1,

$$h(Acc) + (1 - Acc) \log(|\mathcal{A}| - 1) \geq \beta - C.$$

Use the elementary relaxations $h(Acc) \leq 1$ (binary entropy is at most 1) and $\log(|\mathcal{A}| - 1) \leq \log |\mathcal{A}|$ (for $|\mathcal{A}| \geq 2$) to obtain

$$1 + (1 - Acc) \log |\mathcal{A}| \geq \beta - C.$$

Rearrange:

$$1 - Acc \geq \frac{\beta - C - 1}{\log |\mathcal{A}|} \implies Acc \leq 1 - \frac{\beta - C - 1}{\log |\mathcal{A}|}.$$

Because accuracy is trivially at most 1, we write the bound with a cap:

$$Acc \leq \min \left\{ 1, \ 1 - \frac{\beta - C - 1}{\log |\mathcal{A}|} \right\},$$

which is Eq. 4. (When the right-hand side exceeds 1, the $\min\{\cdot, 1\}$ keeps the bound meaningful.)

**Derivation for the Uniform-Distribution Case (Eq. 5).** In the common case where the context does not provide strong cues to distinguish among candidates, the posterior distribution $p(a \mid Q, C)$ over answers $a \in \mathcal{A}$ is close to uniform. Intuitively, this corresponds to situations where many distractor entities of the correct type (e.g., names, dates, or organizations) appear in the context, so that each candidate remains nearly equally plausible given $(Q, C)$. Formally, this means that the entropy of the answer distribution approaches its maximum, i.e.,

$$\beta = H(A \mid Q, C) \approx \log |\mathcal{A}|,$$

since $\log |\mathcal{A}|$ is the entropy of a uniform distribution over $\mathcal{A}$. Equivalently, the KL divergence between $p(a \mid Q, C)$ and the uniform distribution $U(a)$ is small, i.e.,

$$D_{\mathrm{KL}}(p(\cdot \mid Q, C) \,\|\, U(\cdot)) \approx 0,$$

so that the uncertainty is essentially governed by the candidate set size $|\mathcal{A}|$ itself. Since in this regime $\beta \approx \log |\mathcal{A}| \geq \log(|\mathcal{A}| - 1)$, replacing $\log(|\mathcal{A}| - 1)$ by $\beta$ enlarges the left-hand side of the inequality, hence yields a weaker but still valid bound.

Plugging this approximation into Theorem 1 and again relaxing $h(Acc) \leq 1$ gives

$$1 + (1 - Acc)\,\beta \;\geq\; \beta - C.$$

Rearranging to isolate $Acc$:

$$(1 - Acc)\,\beta \;\geq\; \beta - C - 1 \quad \Longrightarrow \quad 1 - Acc \;\geq\; 1 - \frac{C + 1}{\beta} \quad \Longrightarrow \quad Acc \;\leq\; \frac{C + 1}{\beta}.$$

Capping at 1 yields

$$Acc \;\leq\; \min\left\{1, \; \frac{C + 1}{\beta}\right\},$$

which is Eq. 5. This form emphasizes the *capacity–demand ratio* $(C+1)/\beta$ and makes the "accuracy cliff" explicit: the bound equals 1 whenever $\beta \leq C + 1$, and decays hyperbolically once $\beta > C + 1$.

**Remarks.**

- The proof only uses that $\hat{A}$ is a (deterministic) function of $Y$; if $\hat{A}$ were randomized given $Y$, equation 20 would still hold by the data-processing inequality (conditioning on $(Q, C, Y)$ is at least as informative as conditioning on $(Q, C, \hat{A})$).

- The capacity constant $C$ is taken as the *effective* single-pass capacity $H(Y \mid Q, C)$ realized by the decoding policy, but it can be upper-bounded by modeling constraints on $Y$ (e.g., maximum length and vocabulary size).

- Equality conditions in the Fano-style bound are generally not attained in practical settings; the utility of the bound is in predicting the regime change at $\beta \approx C + 1$ and explaining aggregate trends (the "accuracy cliff").

## A.4 DETAILED BENCHMARK CONSTRUCTION

This appendix provides a detailed account of the design principles and generation pipeline for our synthetic, noise-rich Multi-Hop Question Answering (MHQA) benchmark.

### A.4.1 MOTIVATION AND DESIGN PRINCIPLES

As stated in the main text, our primary motivation was to overcome the limitations of existing MHQA datasets, which often lack the fine-grained control over difficulty and the data hygiene necessary for a rigorous evaluation of information-theoretic limits. To this end, our benchmark was designed around three core principles:

1. **Systematic Control over Information Demand ($\beta$):** The benchmark must allow for the precise and independent control of factors known to influence $\beta$, primarily the reasoning hop count ($h$) and the context length ($L$). This enables a systematic study of how performance degrades as information demand scales, allowing for a direct comparison with our theoretical Accuracy Cliff curves.

2. **Resistance to Heuristics and Shortcuts:** The benchmark must be designed to test genuine reasoning rather than retrieval or pattern matching. This is achieved by ensuring all evidence is previously unseen by the model and is embedded within a large number of semantically similar distractors. The high similarity forces the model to perform careful entity disambiguation and information extraction, rather than relying on shallow heuristics.

3. **Maximization of Reasoning Path:** The placement of evidence within the context must enforce a non-trivial reasoning path. A model should not be able to answer a multi-hop question by simply reading the context linearly. Our design forces the model to traverse back and forth across large sections of distractor text, maximizing the cognitive load and testing the model's ability to maintain a coherent reasoning state.

### A.4.2 DATA GENERATION PIPELINE

Our generation pipeline is a programmatic, four-step process designed to instantiate challenging MHQA problems that adhere to the principles above.

**Step 1: Reasoning Chain Instantiation.** We begin by defining a set of abstract semantic templates (e.g., '(Person A, wrote, Book B)', '(Book B, was adapted into, Movie C)'). For a $k$-hop question, we sample $k$ such templates and populate them with distinct entities drawn from a curated knowledge base. This forms the gold evidence chain, $\{e_1, e_2, \ldots, e_k\}$. A question is then programmatically generated to connect the initial entity in $e_1$ to the final entity in $e_k$, with the final entity serving as the ground-truth answer. For example, a 2-hop chain might be '(Frank Herbert, wrote, Dune)' and '(Dune, was adapted into, Dune (2021 film))', leading to the question "What film was adapted from the book written by Frank Herbert?".

**Step 2: Semantically Rich Distractor Generation.** For each piece of gold evidence $e_i$, we generate a set of $N_d$ distractor statements. This is done by taking the semantic template of $e_i$ and substituting its entities with other entities of the same type (e.g., other authors, other books). For instance, for '(Frank Herbert, wrote, Dune)', distractors could be '(Isaac Asimov, wrote, Foundation)' or '(Frank Herbert, wrote, Dune Messiah)'. This process creates a large pool of plausible but factually incorrect statements that are highly similar to the gold evidence, making the task a stringent test of precision.

**Step 3: Context Assembly and Path Maximization.** This step realizes our third design principle. The gold evidence snippets are deliberately placed out of their logical reasoning order within the context. For instance, for a 3-hop task with logical order $e_1 \rightarrow e_2 \rightarrow e_3$, we might place them in the document in the physical order $e_2 \rightarrow e_3 \rightarrow e_1$. The snippets are inserted at regular intervals within the document (e.g., at 1/4, 2/4, and 3/4 of the context length). The generated distractor statements are then randomly shuffled and used to fill the space between the evidence snippets. This strategy forces the model to first find $e_2$ in the middle, use its information to find $e_3$ further down, and then use that result to jump back to near the beginning to find $e_1$.

**Step 4: Noise Padding and Finalization.** Finally, to control the overall context length ($L$), we pad the assembled context with generic, irrelevant noise text (e.g., paragraphs generated by LLMs). This padding is added to the beginning, end, and between existing statements until the target token count (from 500 to 10,000) is reached. This ensures the model must not only handle targeted, similar distractors but also vast amounts of truly irrelevant information, faithfully simulating real-world, noisy long-context scenarios.

This pipeline produces a suite of challenging and controllable datasets, whose key statistics are summarized in Table 1 in the main text. By systematically varying $h$ and $L$, we can precisely map out the performance landscape and validate our theoretical predictions.

---

**Algorithm 1** Multi-hop Reasoning Dataset Construction

---

1: **Input:** $N = 300$, $L = \{500, 1000, \ldots, 10000\}$
2: **Output:** Multi-hop datasets for each target length $L$

3: **Phase 1: Initialize**
4: Define entity dictionary $\mathcal{E}$ with categories (personnel, organizations, etc.)
5: Define templates $\mathcal{T}$; each $t \in \mathcal{T}$ includes entity sequence $E_t$, chain templates $C_t$, questions $Q_t$

6: **Phase 2: Generate Base Chains**
7: **for** $i = 1$ to $N$ **do**
8:     $t \leftarrow \mathcal{T}[i \bmod |\mathcal{T}|]$
9:     $chain_i \leftarrow \text{GENERATECHAIN}(t)$
10: **end for**
11: **function** GENERATECHAIN($t$)
12:     Sample entities for $E_t$ from $\mathcal{E}$
13:     Format $C_t$ with entities to get $chain\_texts$
14:     **return** $(t, chain\_texts, entity\_values)$
15: **end function**

16: **Phase 3: Generate Distractors**
17: **function** GENERATEDISTRACTORS($chain, n_{dist}, n_{var}, n_{noise}$)
18:     Apply distractor templates to create similar and noisy sentences
19:     **return** $(similar, noise)$
20: **end function**

21: **Phase 4: Build Multi-length Dataset**
22: **for** $L_i \in L$ **do**
23:     Compute $n_{dist}, n_{noise}$ from $L_i$
24:     **for** each $chain_i$ **do**
25:         **for** $h = 1$ to $4$ **do**
26:             Scale distractors: $n_{dist}^{(h)} \leftarrow \lfloor n_{dist} \cdot (1 + 0.6(h-1)) \rfloor$
27:             $distractors \leftarrow \text{GENERATEDISTRACTORS}(chain_i, n_{dist}^{(h)}, 5, n_{noise})$
28:             $sample \leftarrow \text{BUILDSAMPLE}(h, chain_i, distractors)$
29:         **end for**
30:     **end for**
31: **end for**
32: **function** BUILDSAMPLE($h, chain, D$)
33:     $q \leftarrow Q_{chain.template}[h-1]$; $a \leftarrow chain.entities[h]$
34:     $S \leftarrow$ first $h$ sentences from $chain.chain\_texts$
35:     $ctx \leftarrow \text{CREATECONTEXT}(S, D)$
36:     **return** $(q, a, S, D, ctx)$
37: **end function**

38: **Phase 5: Assemble Context**
39: **function** CREATECONTEXT($S, D$)
40:     Interleave $D.similar$ and $D.noise$
41:     Insert $S$ at fixed positions based on $h$ (e.g. for $h = 3$: positions $[1/4, 2/4, 3/4]$)
42:     Pad if needed to target token length
43:     **return** context
44: **end function**

45: **Phase 6: Save Output**
46: **for** $h \in \{1, 2, 3, 4\}$, $L_i \in L$ **do**
47:     Write all $(q, a, ctx)$ triples to `$h hop/multi_hop_chain_$Lk.json`
48: **end for**
49: Save dataset statistics

A.5 Fitting Algorithm

This section details the procedure used to fit the relationship between *effective information demand* and task performance (F1). We formalize the parametric model, the loss function, the search strategy, numerical safeguards, and the computational complexity, and we provide pseudocode for reproducibility.

**Model Assumption (Beta–Bound Structure)**   For a given *reasoning depth* $h \in \{1, 2, 3, 4\}$ and *context length* $L$ (in tokens), we posit that the effective information demand is

$$\beta(h, L) = \alpha L \gamma^{h-1} + \beta_0,$$

with parameters $\alpha > 0$, $\gamma > 1$, and $\beta_0 \geq 0$. The attainable F1 is upper-bounded by an inverse dependence on $\beta$:

$$\widehat{F1}(h, L) = \min\left(1, \frac{C+1}{\beta(h, L)}\right),$$

where $C \geq 0$ captures a constant-information offset and induces a kink at $\widehat{F1} = 1$.

**Objective**   Given empirical observations $F1_{\text{emp}}(h, L)$, we estimate $(\alpha, \gamma, \beta_0, C)$ by minimizing the mean absolute error (MAE):

$$\mathcal{L}(\alpha, \gamma, \beta_0, C) = \frac{1}{N} \sum_{(h,L)} \left|\widehat{F1}(h, L) - F1_{\text{emp}}(h, L)\right|,$$

where $N$ is the number of $(h, L)$ pairs (here $N = 4 \times 6 = 24$).

**Search Strategy: Fine-Grained Grid Search**   To avoid local minima introduced by the non-smooth kink at $\widehat{F1} = 1$, we employ a *fine-grained grid search* over

$$\alpha \in \mathcal{A}, \quad \gamma \in \mathcal{G}, \quad \beta_0 \in \mathcal{B}, \quad C \in \mathcal{C}.$$

Unless otherwise stated, we use

$$\mathcal{A} = \text{logspace}\left(10^{-4}, 10^{-2}, 15\right), \quad \mathcal{G} = \text{linspace}\left(1.05, 3.00, 20\right),$$

$$\mathcal{B} = \text{linspace}\left(0, 200, 21\right), \quad \mathcal{C} = \text{linspace}\left(20, 400, 25\right).$$

Each method (Direct, CoT, S-R, S-C, ReAct, P&S, S-A) is fitted independently, yielding its own $(\alpha, \gamma, \beta_0, C)$.

**Implementation Details and Numerical Stability**

- **Vectorization.** For each $(\alpha, \gamma)$ pair, we first compute the base term $\alpha L \gamma^{h-1}$ for all $(h, L)$, and then sweep over $\beta_0$ and $C$. This reduces redundant computation and improves throughput.
- **Stability at small $\beta$.** We enforce $\beta(h, L) \leftarrow \max\{\beta(h, L), 10^{-9}\}$ to avoid division by zero.
- **Upper-bound consistency.** The cap $\min(1, \cdot)$ ensures fidelity in the high-resource regime where $\widehat{F1} \to 1$.
- **Optional weighting.** If desired, a weight $w(h, L)$ can be introduced in $\mathcal{L}$ to emphasize specific depths or lengths (default: uniform).

**Computational Complexity**   Let $|\mathcal{A}|$, $|\mathcal{G}|$, $|\mathcal{B}|$, and $|\mathcal{C}|$ denote the grid sizes and $N$ the number of samples. The complexity per method is

$$\mathcal{O}\left(|\mathcal{A}| \, |\mathcal{G}| \, |\mathcal{B}| \, |\mathcal{C}| \, N\right).$$

Since $N = 24$ is small, the overall runtime remains practical under vectorized implementations. Faster variants can be obtained via coarse-to-fine (multistage) search or by shrinking grid ranges.

---

**Algorithm 2** Fine-Grained Grid Fitting for One Method

---

**Require:** Data $\{(h_i, L_i, \text{F1}_i)\}_{i=1}^N$; grids $\mathcal{A}, \mathcal{G}, \mathcal{B}, \mathcal{C}$
1: best_loss $\leftarrow +\infty$, best $\leftarrow \varnothing$
2: **for** $\alpha \in \mathcal{A}$ **do**
3:     **for** $\gamma \in \mathcal{G}$ **do**
4:         $\text{base}_i \leftarrow \alpha L_i \gamma^{h_i - 1} \ \ \forall i$
5:         **for** $\beta_0 \in \mathcal{B}$ **do**
6:             $\beta_i \leftarrow \max(\text{base}_i + \beta_0, 10^{-9})$
7:             **for** $C \in \mathcal{C}$ **do**
8:                 $\widehat{\text{F1}}_i \leftarrow \min\big(1, (C+1)/\beta_i\big)$
9:                 $\text{loss} \leftarrow \frac{1}{N}\sum_i \big|\widehat{\text{F1}}_i - \text{F1}_i\big|$
10:                 **if** loss $<$ best_loss **then**
11:                     best_loss $\leftarrow$ loss; best $\leftarrow (\alpha, \gamma, \beta_0, C)$
12:                 **end if**
13:             **end for**
14:         **end for**
15:     **end for**
16: **end for**
17: **return** best, best_loss

---

**Reproducibility** All methods share the same $(h, L)$ grid with $h \in \{1, 2, 3, 4\}$ and $L \in \{0.5k, 1k, 2k, 4k, 8k, 10k\}$. We expand this grid with `meshgrid(indexing=``ij'')` and flatten to length $N = 24$ vectors for fitting. The default metric is MAE; alternative choices (e.g., MAPE or weighted MAE) produce qualitatively similar trends. Parameter uncertainty can be assessed via bootstrap resampling over $(h, L)$ pairs.

## A.6 QWEN3-14B FITTING PARAMETERS

As shown in Figure 3, the plug-in bound provides an excellent global fit, as reflected in the low MAE across all methods, and it captures method-level differences through the parameters $(\gamma, C, \beta_0)$. CoT and S-C expand the usable regime by increasing $C$ and reducing $\gamma$, thereby mitigating the cliff. S-A incurs a large base-demand penalty ($\beta_0$), which counteracts the benefit of a higher $C$. Removing distractors nearly eliminates hop inflation ($\gamma \approx 1$), indicating that compounding arises primarily from noise rather than depth. Taken together, these results empirically substantiate the *accuracy cliff* predicted by our theory.

Table 3: Fitted parameters of the plug-in accuracy bound (MAE minimization) of Qwen3-14B. Larger $C$ indicates higher effective single-pass capacity; smaller $\gamma$ indicates weaker hop inflation.

| Method | $\alpha$ | $\gamma$ | $\beta_0$ | $C$ | MAE |
|--------|----------|----------|-----------|-----|-----|
| Direct | 0.0100 | 3.000 | 40 | 67.5 | 0.0963 |
| CoT | 0.0100 | 2.076 | 0 | 131 | 0.0320 |
| S-C | 0.00720 | 2.076 | 0 | 99.2 | 0.0273 |
| S-R | 0.0100 | 2.282 | 110 | 147 | 0.0531 |
| ReAct | 0.0100 | 1.768 | 80 | 115 | 0.0429 |
| P&S | 0.00268 | 1.974 | 20 | 35.8 | 0.0444 |
| S-A | 0.00518 | 1.974 | 160 | 162 | 0.0747 |
| w/o D. | 0.0100 | 1.050 | 70 | 67.5 | 0.0589 |

Table 4: Fitted parameters of the plug-in accuracy bound (MAE minimization) of Qwen3-8B. Larger $C$ indicates higher effective single-pass capacity; smaller $\gamma$ indicates weaker hop inflation.

| Method | $\alpha$ | $\gamma$ | $\beta_0$ | $C$ | MAE |
|--------|----------|----------|-----------|-----|-----|
| Direct | 0.00720 | 3.000 | 10 | 20 | 0.1061 |
| CoT | 0.0100 | 2.076 | 60 | 131 | 0.0426 |
| S-C | 0.0100 | 2.076 | 60 | 131 | 0.0343 |
| S-R | 0.00518 | 2.076 | 50 | 51.7 | 0.0747 |
| ReAct | 0.00518 | 2.076 | 70 | 83.3 | 0.0465 |
| P&S | 0.0100 | 2.487 | 160 | 178 | 0.0480 |
| S-A | 0.00373 | 2.795 | 130 | 131 | 0.0475 |

## A.7 EXPERIMENTAL RESULTS OF QWEN3-8B

**Theory fit for single-pass methods.** The plug-in accuracy bound in Eq. 7 fits Qwen3–8B's single-pass baselines well (Table 3, Figure 6). *Direct* exhibits a small effective per-pass capacity ($C \approx 20$) and strong hop inflation ($\gamma = 3.0$), hence an early accuracy cliff. *CoT* and *S-C* attain a much larger capacity ($C \approx 131$) with moderate hop inflation ($\gamma \approx 2.08$) and the lowest MAE (0.0426/0.0343), so their empirical points hug the theoretical envelope longer. *P&S* shows the largest $C$ ($\approx 178$) but also a large base demand ($\beta_0 = 160$) and higher $\gamma$ ($\approx 2.49$), which offsets its capacity at greater depth/length. *ReAct* and *S–R* have mid-range capacities ($C \approx 83.3$ and 51.7) and degrade earlier. *S–A* has sizable $\beta_0$ (130) and high $\gamma$ ($\approx 2.80$), reflecting method-specific overheads that accelerate the cliff despite a decent $C$. Overall, the fitted overlays confirm the accuracy-cliff picture: empirical F1 follows the bound and collapses once the fitted demand $\beta$ crosses $C+1$.

**Performance of InfoQA.** *1. Depth robustness.* On 2–4 hops, InfoQA overall average is **0.74** vs. **0.66** for S–C and **0.65** for CoT. By hop: at 2-hop, **0.89** (InfoQA) vs. 0.82 (S–C); at 3-hop, **0.64** vs. 0.63 (S–C/ ReAct); at 4-hop, **0.68** vs. 0.52 (S–C). Gains grow with depth: at 4-hop and long contexts (e.g., 8k), InfoQA reaches **0.67**, while the best single-pass baseline tops out around **0.16**. This matches the theory: capacity-aware decomposition keeps each step's demand $\beta_k \le C$. *2. Length robustness.* InfoQA maintains strong performance as context length increases. For 2-hop at 8k tokens, it achieves **0.74** (vs. 0.67 for S–A and 0.57 for S–C); for 3-hop at 10k tokens, it reaches **0.28**, exceeding the best single-pass alternative (0.21 for S–C). Even at 1-hop, where all methods are strong, InfoQA remains competitive (average **0.93**) without relying on a single long reasoning trace.

Table 5: Qwen3-8B's Average F1 scores across different reasoning depths and context lengths. We compare InfoQA with single-pass baselines: Chain-of-Thought (CoT), Self-Refine (S-R), Self-Consistency (S-C), ReAct, Plan-and-Solve (P&S), Self-Ask (S-A).

| Hops | Context Length | Average F1 Score | | | | | | | |
|---|---|---|---|---|---|---|---|---|---|
| | | Direct | CoT | S-R | S-C | ReAct | P&S | S-A | InfoQA |
| **1** | 0.5k | 0.98 | **1.00** | 0.96 | **1.00** | **1.00** | **1.00** | 0.98 | **1.00** |
| | 1k | 0.98 | **1.00** | 0.94 | **1.00** | 0.98 | **1.00** | 0.98 | **1.00** |
| | 2k | 0.94 | 0.99 | 0.93 | 0.99 | 0.96 | 0.97 | 0.96 | **1.00** |
| | 4k | 0.91 | **0.97** | 0.81 | **0.97** | 0.93 | 0.90 | 0.91 | **0.97** |
| | 8k | 0.83 | **0.91** | 0.58 | **0.91** | 0.76 | 0.78 | 0.82 | 0.82 |
| | 10k | 0.76 | 0.87 | 0.49 | **0.89** | 0.74 | 0.68 | 0.71 | 0.78 |
| **2** | 0.5k | 0.57 | **1.00** | 0.96 | **1.00** | 0.98 | 0.99 | 0.89 | **1.00** |
| | 1k | 0.44 | **0.99** | 0.88 | **0.99** | 0.96 | 0.97 | 0.77 | 0.96 |
| | 2k | 0.28 | 0.95 | 0.71 | 0.94 | 0.93 | 0.87 | 0.89 | **0.98** |
| | 4k | 0.23 | 0.93 | 0.43 | 0.95 | 0.49 | 0.78 | 0.78 | **0.96** |
| | 8k | 0.10 | 0.54 | 0.15 | 0.57 | 0.49 | 0.51 | 0.67 | **0.74** |
| | 10k | 0.10 | 0.47 | 0.56 | 0.48 | 0.44 | 0.42 | 0.66 | **0.72** |
| **3** | 0.5k | 0.50 | 0.96 | 0.86 | **0.97** | 0.96 | 0.83 | 0.94 | 0.92 |
| | 1k | 0.25 | 0.89 | 0.64 | 0.91 | 0.88 | 0.69 | 0.84 | **0.93** |
| | 2k | 0.13 | 0.79 | 0.51 | 0.81 | 0.82 | 0.62 | 0.67 | **0.83** |
| | 4k | 0.09 | 0.60 | 0.27 | 0.58 | **0.62** | 0.42 | 0.51 | 0.61 |
| | 8k | 0.04 | **0.32** | 0.17 | 0.31 | 0.27 | 0.20 | 0.25 | 0.28 |
| | 10k | 0.02 | 0.20 | 0.10 | 0.21 | 0.15 | 0.12 | 0.10 | **0.28** |
| **4** | 0.5k | 0.12 | 0.94 | 0.84 | 0.97 | 0.91 | 0.89 | 0.70 | **1.00** |
| | 1k | 0.09 | 0.84 | 0.72 | 0.88 | 0.77 | 0.72 | 0.63 | **0.96** |
| | 2k | 0.03 | 0.66 | 0.49 | 0.64 | 0.56 | 0.45 | 0.47 | **0.70** |
| | 4k | 0.01 | 0.44 | 0.29 | 0.40 | 0.32 | 0.28 | 0.32 | **0.63** |
| | 8k | 0.00 | 0.15 | 0.12 | 0.16 | 0.09 | 0.08 | 0.16 | **0.67** |
| | 10k | 0.00 | 0.09 | 0.05 | 0.09 | 0.03 | 0.04 | 0.10 | **0.12** |
| **Overall Average (2–4 hop)** | | 0.17 | 0.65 | 0.49 | 0.66 | 0.59 | 0.55 | 0.58 | **0.74** |
| | **1 hop Average** | 0.90 | 0.96 | 0.79 | **0.96** | 0.90 | 0.89 | 0.89 | 0.93 |
| | **2 hop Average** | 0.29 | 0.81 | 0.62 | 0.82 | 0.72 | 0.76 | 0.78 | **0.89** |
| | **3 hop Average** | 0.17 | 0.63 | 0.42 | 0.63 | 0.62 | 0.48 | 0.55 | **0.64** |
| | **4 hop Average** | 0.04 | 0.52 | 0.42 | 0.52 | 0.45 | 0.41 | 0.40 | **0.68** |
| | | Context Average (2–4 hop) | | | | | | | |
| | 0.5k | 0.40 | 0.97 | 0.89 | **0.98** | 0.95 | 0.90 | 0.84 | 0.97 |
| | 1k | 0.26 | 0.91 | 0.75 | 0.93 | 0.87 | 0.79 | 0.75 | **0.95** |
| | 2k | 0.15 | 0.80 | 0.57 | 0.80 | 0.77 | 0.65 | 0.68 | **0.84** |
| | 4k | 0.11 | 0.66 | 0.33 | 0.64 | 0.48 | 0.49 | 0.54 | **0.73** |
| | 8k | 0.05 | 0.34 | 0.15 | 0.35 | 0.28 | 0.26 | 0.36 | **0.56** |
| | 10k | 0.04 | 0.25 | 0.24 | 0.26 | 0.21 | 0.19 | 0.29 | **0.37** |

Figure 6: Qwen3-8B's Empirical F1 vs. theoretical curves across single-pass methods. The x-axis shows the estimated effective information demand ($\beta$), fitted per method, and the y-axis shows the F1 score.

