# OpenReview forum: "A Fano-Style Accuracy Upper Bound for LLM Single-Pass Reasoning in Multi-Hop QA"
_ICLR.cc/2026/Conference — ICLR 2026 Poster_

### Official Review · Reviewer_KUVW · 2025-10-20

**Soundness:** 2
**Presentation:** 2
**Contribution:** 2
**Rating:** 4
**Confidence:** 3

**Summary:**

The paper addresses the challenge of Multi-Hop Question Answering (MHQA), where multiple, dispersed, and interdependent pieces of evidence need to be integrated through sequential reasoning. This task is particularly difficult for large language models (LLMs) due to their finite output capacity per pass, which makes the integration of relevant evidence unreliable once the task complexity exceeds model capacity. The paper formalizes this bottleneck by introducing a Fano-style accuracy upper bound, which defines a theoretical performance ceiling for single-pass LLMs. This bound shows that as task complexity increases beyond model capacity, accuracy inevitably collapses.

Building upon this theoretical foundation, the authors propose a new multi-call framework for MHQA, called InfoQA. InfoQA addresses the capacity overflow by ensuring high per-step accuracy through capacity-aware task decomposition and active pruning of prior reasoning traces. This helps keep the information load within the model’s single-pass limit. Furthermore, InfoQA enhances robustness through a dependency-explicit workflow, which allows for precise control over the reasoning path. To validate their approach, the authors construct a stringent, noise-rich benchmark and show that the model's performance aligns with the theoretical capacity curves. Experimental results demonstrate that InfoQA consistently improves performance in MHQA tasks.

**Strengths:**

1. The introduction of the Fano-style accuracy upper bound provides a unique theoretical foundation for understanding the limitations of single-pass LLMs in handling complex MHQA tasks. This may provide some insights for future research in capacity-aware representation for LLMs.

2. The work provides a comprehensive experimental result which involves various settings.

**Weaknesses:**

1. While the InfoQA framework offers clear benefits, the multi-call nature of the approach could introduce overhead, especially for tasks requiring many iterations. This might limit its scalability for more real-time applications or systems with stricter performance requirements.

2. Overemphasis on Theoretical Bound: The theoretical Fano-style upper bound offers valuable insights into the performance limits of single-pass LLMs. However, the heavy focus on this theoretical bound may overshadow more practical solutions for real-world MHQA scenarios. The boundary established may not always align with the practical constraints of large-scale systems, where the focus should be on scalable, efficient reasoning. Moreover, while the paper introduces many complex formulas, they are not always effectively integrated with the core problem of MHQA. The formulas provide theoretical rigor but may not offer direct, actionable solutions for improving the model’s real-world performance.

3. It could benefit from a more direct comparison with other multi-hop question answering systems or frameworks.

**Questions:**

Authors may consider revising the presentation and add a main figure to help readers intuitively understand the paper and more baselines are expected to validate the effectiveness of this framework.

---

> ### Author Response · Authors · 2025-11-19
> **Comment 1**
>
> W1: While the InfoQA framework offers clear benefits, the multi-call nature of the approach could introduce overhead, especially for tasks requiring many iterations. This might limit its scalability for more real-time applications or systems with stricter performance requirements.
>
> A1: We sincerely appreciate this practical observation. We fully acknowledge that the multi-call paradigm inherently introduces latency and computational overhead (in terms of time and token count) compared to single-pass methods. However, this trade-off is often a theoretical necessity rather than a design flaw.
>
> As our analysis of the "Accuracy Cliff" demonstrates, when a task's information demand ($\beta$) exceeds the model's single-pass capacity ($C$), the single-pass paradigm is fundamentally untenable—it does not just degrade; it collapses. In such cases, the multi-call overhead is the requisite computational cost to bridge this capacity gap and achieve correctness.
>
> This approach becomes indispensable in two highly relevant scenarios where capability is prioritized over real-time latency. (1) Extremely Complex Tasks: For tasks such as analyzing extensive financial reports or mapping complex corporate relationships, the complexity ($\beta$) often exceeds the single-pass capacity of even the strongest state-of-the-art closed-source LLMs. In these cases, multi-call is the only viable solution to complete the task reliably. (2) Resource-Constrained Environments: In scenarios requiring local deployment for privacy (e.g., processing sensitive data), users are often restricted to smaller models with lower intrinsic output capacity. The multi-call paradigm enables these "weaker" models to successfully solve complex tasks that would otherwise require a much larger server-side model.
>
> Additionally, we do not intend to claim that multi-call is universally superior. For scenarios requiring instant feedback where $\beta < C$, single-pass remains the optimal choice. Our contribution is providing a diagnostic perspective based on information capacity, enabling researchers and practitioners to analyze the "Supply vs. Demand" of their specific application and choose the appropriate paradigm accordingly.
>
> ---
> W3: It could benefit from a more direct comparison with other multi-hop question answering systems or frameworks.
>
> A3: We sincerely appreciate this suggestion. We would like to clarify that our baseline selection was a deliberate methodological strategy strictly aligned with our research scope and intended theoretical contribution.
>
> The primary objective of this work is to analyze the capacity constraints of LLMs from an information-theoretic perspective and to verify the "Accuracy Cliff" phenomenon in the single-pass paradigm. Consequently, a direct comparison with single-pass baselines is scientifically the most relevant approach to rigorously validate our core conclusion: single-pass methods inevitably fail when the task's information demand $\beta$ exceeds the model capacity $C$, whereas the multi-call paradigm (exemplified by InfoQA) remains robust. In contrast, a broad comparison with heterogeneous engineering systems might obscure this clear theoretical signal.
>
> Nevertheless, the suggestion is valuable and forward-looking. We fully agree that comparing with a broader range of frameworks is the critical next step for this line of research. Now that our work has established the necessary theoretical guarantee for the multi-call paradigm, we are eager to follow your recommendation in our future work to expand InfoQA into a more comprehensive system and conduct extensive benchmarking against diverse frameworks across various datasets to further demonstrate its practical superiority. Thank you again for helping us clarify this important roadmap.
>
> ---
> Q1: q1: Authors may consider revising the presentation and add a main figure to help readers intuitively understand the paper and q2: more baselines are expected to validate the effectiveness of this framework.
>
> a1: We sincerely appreciate this suggestion to improve the visual presentation. To help readers intuitively grasp the core concepts, we have incorporated clearer schematic illustrations and textual explanations into the revised Figure 1 to better illustrate the resulting "capacity overflow." Additionally, we have improved Figure 2 by adding explicit annotations for the model's capacity ($C$) and clarifying the critical relationship between task demand ($\beta$) and accuracy. We hope these updated figures now can serve as an intuitive visual guide to our theoretical framework.
>
> a2: Regarding the baselines, please kindly refer to our response to W3, where we discuss in detail how our baseline selection was a deliberate methodological strategy strictly aligned with our theoretical research scope.

---

> ### Author Response · Authors · 2025-11-19
> **Comment 2**
>
> W2: Overemphasis on Theoretical Bound: The theoretical Fano-style upper bound offers valuable insights into the performance limits of single-pass LLMs. However, the heavy focus on this theoretical bound may overshadow more practical solutions for real-world MHQA scenarios. The boundary established may not always align with the practical constraints of large-scale systems, where the focus should be on scalable, efficient reasoning. Moreover, while the paper introduces many complex formulas, they are not always effectively integrated with the core problem of MHQA. The formulas provide theoretical rigor but may not offer direct, actionable solutions for improving the model’s real-world performance.
>
> A2: We deeply appreciate this perspective and fully agree that theory must ultimately serve practice. However, we suggest that our theoretical framework is intended not as abstract rigor for its own sake, but as a necessary diagnostic foundation that enables the design of effective, actionable solutions.
>
> First, we believe that accurately identifying the fundamental root causes of a problem is a prerequisite for proposing effective solutions. The formulas in Section 2 (Theorem 1) serve to diagnose the root cause of failure: the "Accuracy Cliff." They quantify exactly why single-pass methods fail—not because of bad prompt engineering, but because the task's information demand ($\beta$) mathematically exceeds the model's channel capacity ($C$). Section 3 explicitly integrates these formulas with the "core problem of MHQA" by formalizing the twin challenges of Stepwise Capacity Overflow (Eq. 6 & 7) and Cross-Step Error Accumulation (Eq. 10).
>
> Far from overshadowing practical solutions, we hope to demonstrate that **the theoretical bound directly dictated the engineering design of our InfoQA framework**, meaning that every component of InfoQA is an "actionable solution" derived specifically from our formulas. For instance, since Equation 6 shows that demand $\beta$ grows super-linearly with hop count $h$ and context length $L$, we counter the growth of $h$ via "Capacity-Aware Decomposition" to try to make the per-step demand $\beta_k$ stay below capacity, and we address the growth of $L$ through "Iterative Query Contraction" to actively prune the context. The consistent performance improvements demonstrated in Table 2 suggest that these theoretically derived actions can directly translate into real-world gains.
>
> Finally, regarding the valid concern that the boundary does not immediately solve scalability issues, we draw deep inspiration from the philosophy of Claude Shannon, whose Information Theory [1] laid the cornerstone of the modern information technology age and underpins the very existence of Artificial Intelligence. While we certainly do not dare to compare our humble contributions to his monumental legacy, we strive to follow the methodological path he illuminated: defining theoretical limits is often the necessary precursor to engineering mastery. Just as the Channel Capacity limit was established long before efficient coding schemes were engineered to approach it, our modest aspiration is to establish a "Reasoning Capacity" limit for LLMs. We sincerely hope this theoretical groundwork can serve as a helpful guide for future research into scalable system design.
>
> We respectfully suggest that the practical utility of such a limit is profound. Without this theoretical boundary, engineers might inadvertently expend vast resources trying to optimize single-pass inference for tasks that are mathematically impossible to solve within that paradigm (where $\beta > C$). By defining "what is impossible" through this theoretical bound, we hope to guide the community toward "what is possible"—the multi-call paradigm—providing a necessary theoretical foundation for future scalable system design.
>
> [1] Shannon, Claude E. "A mathematical theory of communication." The Bell system technical journal 27.3 (1948): 379-423.
>
> ---
> In conclusion, we sincerely thank you for the thoughtful and practical feedback. We have updated Figure 1 and Figure 2 in the revised manuscript to enhance clarity and intuitiveness as per your suggestion. We hope that our responses and revisions help to clarify our work and address your concerns. If so, we would be deeply grateful if you could kindly consider raising the evaluation.

---

> ### Author Response · Authors · 2025-11-24
> **A respectful note on our response**
>
> Dear Reviewer KUVW,
>
> We respectfully hope that our response helps to clarify your concerns. We remain fully at your disposal should you need any further clarification.
>
> Best regards,
>
> The Authors

---

> > ### Author Response · Authors · 2025-11-28
> >
> > Dear Reviewer KUVW,
> >
> >
> > As the discussion period is drawing to a close, we wanted to briefly check if our previous response and the revised manuscript have sufficiently addressed your concerns, particularly regarding the trade-off between multi-call overhead and reliability (W1) and the practical utility of our theoretical framework (W2).
> >
> > We fully recognize your point that theoretical bounds must translate into practice. We hope our clarification on how the "Accuracy Cliff" serves as a necessary diagnostic tool for system design provides the context you were looking for.
> >
> > We remain eager to answer any final questions you might have to help reconsider your assessment.
> >
> >
> > Best regards,
> >
> > The Authors

---

### Official Review · Reviewer_DhBc · 2025-10-27

**Soundness:** 3
**Presentation:** 4
**Contribution:** 3
**Rating:** 6
**Confidence:** 3

**Summary:**

This paper utilizes Information Theory to analyze the capability limitations of the single-pass solution to multi-hop question answering (MHQA), reveals the "Accuracy Cliff" phenomenon, identifies two challenges ("Stepwise Capacity Overflow" and "Cross-Step Error Accumulation"), and then proposes a multi-call framework (InfoQA) for MHQA.

**Strengths:**

1. The analysis, grounded in Information Theory, is insightful, and it reveals the limitations of single-pass MHQA solutions.
2. The experimental results (Table 2) demonstrate the effectiveness of the proposed InfoQA method.

**Weaknesses:**

1. I appreciate the theoretical analysis and experimental findings in this work, but I have some concerns regarding the analysis based on Information Theory:
    - I do not think information theory is suitable to assess the capability of natural language. Even if we consider all the possible values of a token (i.e., the size of the vocabulary) and compute the entropy or self-information of this token $H(X) = - \sum_{x \in X} p(x) log p(x)$, this still ignores the variability, richness, and ambiguity of the meaning conveyed by natural language. A single word can mean a lot.
    - With a finite number of tokens in a single pass, inefficient reasoners may suffer from the limited output budget, but efficient ones can use fewer tokens to achieve desirable results.
    - In addition, inefficient reasoners can utilize inference-time scaling strategies to extend their single-pass reasoning trace, making the limited budget not much of a fundamental issue.
2. The experiments are conducted on the synthesized dataset proposed by this paper, without empirical results on existing widely used MHQA benchmarks.
3. Section 5.2 and Figure 5: The empirical results do not align well with the theoretical curves (Accuracy Cliff) to me.

**Questions:**

- Note: I would rate this paper as 7 initially, although this option is not available.

---

> ### Author Response · Authors · 2025-11-19
> **Comment 1**
>
> W1.1: I do not think information theory is suitable to assess the capability of natural language. Even if we consider all the possible values of a token (i.e., the size of the vocabulary) and compute the entropy or self-information of this token, this still ignores the variability, richness, and ambiguity of the meaning conveyed by natural language. A single word can mean a lot.
>
> A1.1: Thank you for this insightful comment. We fully agree that natural language possesses variability, richness, and ambiguity that extend far beyond simple statistical entropy. A single token can indeed carry profound semantic weight. However, we would like to clarify that our framework does not attempt to measure the semantic complexity of natural language in the abstract, but rather the transmission capacity of the LLM's physical representation of that language. While Information Theory may not capture the full nuance of human semantics, it is the precise mathematical tool for analyzing the operational reality of an LLM. Fundamentally, LLMs do not process "meanings"; they process discrete tokens, which are implemented as matrices and ultimately stored and computed as digital bits (0s and 1s).
>
> **A Bottom-Up Perspective (From Bits to Tokens):** Our analysis treats the LLM as a noisy channel to establish a theoretical boundary applicable to this machine representation. In this context, $H(Y)$ does not quantify the "meaning" of the output, but rather the theoretical upper limit of the information budget available in the output sequence to encode that meaning.
>
> **Capturing "Richness and Ambiguity":** Furthermore, the "richness and ambiguity" you mentioned are mathematically captured in our framework by the task's information demand, $\beta \triangleq H(A|Q,C)$. If a task involves ambiguous language where "a single word means a lot," the uncertainty regarding the precise answer given the context increases. This leads to a higher $\beta$.
>
> **Core Conclusion: The Capacity Bottleneck.** Our contribution (Theorem 1) reveals a bottleneck: no matter how semantically rich the language is, the model must output the answer through a finite sequence of tokens. If the information required to resolve the ambiguity ($\beta$) exceeds the maximum information this discrete output sequence can physically encode ($C \triangleq H(Y)$), the transmission of the correct answer becomes mathematically impossible.
>
> **Summary: The Role of Information Theory.** Thus, Information Theory serves here as a necessary bounding tool for the machine's capacity to represent language, explaining why performance collapses when the "container" ($C$) is too small for the "content" ($\beta$), regardless of the semantic nature of that content.
>
> ---
> W1.2: With a finite number of tokens in a single pass, inefficient reasoners may suffer from the limited output budget, but efficient ones can use fewer tokens to achieve desirable results.
>
> A1.2：Thank you for these thoughtful questions regarding the boundaries of our theory. We agree that efficient reasoners can indeed encode information more densely, effectively achieving a higher utilization of the channel capacity. Our framework explicitly accounts for this. As shown in Table 4, "efficient" methods like CoT fit a significantly higher effective capacity ($C \approx 131$) compared to Direct Prompting ($C \approx 20$).
>
> However, even the most efficient reasoner is bound by the theoretical maximum entropy of the output space ($H(Y)$). As discussed in A1.1, since each generated token has a finite information representation capacity defined by the vocabulary size and embedding space, the sequence has a hard information ceiling. There is a thermodynamic limit to how much information can be compressed into a token sequence. Once the task's information demand $\beta$ exceeds this physical limit, no amount of efficiency can prevent information loss.

---

> ### Author Response · Authors · 2025-11-19
> **Comment 2**
>
> W1.3: In addition, inefficient reasoners can utilize inference-time scaling strategies to extend their single-pass reasoning trace, making the limited budget not much of a fundamental issue.
>
> A1.3: Regarding inference-time scaling, while adding reasoning tokens helps the model approach its theoretical capacity limit, it cannot extend it indefinitely due to the bottleneck of finite representation. Each subsequently generated token has a fixed number of parameters (i.e., the fixed dimensionality of the model's embedding space). As the reasoning chain extends, the current token must essentially "compress" the relevant information from an ever-growing preceding context into this fixed-size vector to pass it to the next step.
>
> Consequently, there is a theoretical limit to how much historical information a new token can represent. Once the information load exceeds this representational limit, the new tokens can no longer "hold" the necessary prior information, causing subsequent reasoning to fail. This establishes the hard capacity upper bound we analyze. Meanwhile, as modeled in Equation 6, the task difficulty grows super-linearly. The "scaling" strategy fails to catch up with this exponential demand due to the representation bottleneck mentioned above, and the accumulated intermediate tokens also act as noise, blocking the attention mechanism from effectively accessing the initial evidence.
>
> Our work analyzes the limitations on the model side (Supply $C$) and the requirements on the task side (Demand $\beta$). We argue that the path to solving complex reasoning tasks must involve simultaneous efforts in two directions: 1) increasing the model's reasoning capacity and 2) researching how to better decompose tasks to reduce their information demand, as demonstrated by our framework.
>
> ---
> W2: The experiments are conducted on the synthesized dataset proposed by this paper, without empirical results on existing widely used MHQA benchmarks.
>
> A2: Thank you for raising this point regarding dataset selection. This was a deliberate methodological choice driven by the following factors.
>
> First, as discussed in Section 5.1, our primary objective is to rigorously test the "Accuracy Cliff" predicted by our information-theoretic bounds. This requires the ability to systematically control the information demand ($\beta$) by independently varying the reasoning depth (hop count) and noise levels (context length). Since existing benchmarks are typically constructed via crowd-sourcing (as in HotpotQA) or bottom-up composition over fixed corpora (as in MuSiQue), they inherently lack the structural mechanism to decouple these variables, making it impossible to isolate them to verify the theory.
>
> Second, widely used benchmarks often suffer from "data contamination," "shortcut learning," and potential "knowledge leakage" (where models memorize answers during pre-training). Our synthetic benchmark is designed to be more stringent: it uses fictitious entities to prevent memorization and uses high semantic similarity between evidence and distractors to enforce genuine multi-hop reasoning rather than heuristic guessing. This ensures that the observed performance drops are strictly due to the capacity bottleneck we aim to prove.
>
> ---
> W3: Section 5.2 and Figure 5: The empirical results do not align well with the theoretical curves (Accuracy Cliff) to me.
>
> A3: We respectfully suggest that the empirical results exhibit a strong alignment with the theoretical predictions regarding the overall trend.
>
> As shown in Figures 5 and 6, the data clearly demonstrate the distinct "Accuracy Cliff" phenomenon predicted by our theory: a high-performance plateau in the capacity-sufficient regime, followed by a sharp, hyperbolic collapse once the information demand exceeds the capacity threshold. The low Mean Absolute Error (MAE) values reported in Appendix A.6 (Table 3) further quantify this fit.
>
> However, we acknowledge the deviations between specific data points and the curve. As you insightfully noted in Q1, natural language semantics are not perfectly equivalent to bit-level information. The inherent variability and richness of natural language introduce complexity that a pure information-theoretic bound abstracts as a unified variable. This inevitably leads to some empirical variance (error) around the theoretical limit. Nevertheless, the fact that the macroscopic "Cliff" behavior remains robust despite this semantic gap strongly validates the existence of the capacity bottleneck.
>
> ---
> In conclusion, we sincerely thank you for your profound and constructive feedback. We hope that our responses help to clarify our work and address your concerns. If so, we would be deeply grateful if you could kindly consider raising the evaluation.

---

> ### Author Response · Authors · 2025-11-24
> **A respectful note on our response**
>
> Dear Reviewer DhBc,
>
> We respectfully hope that our response helps to clarify your concerns. We remain fully at your disposal should you need any further clarification.
>
> Best regards,
>
> The Authors

---

> ### Comment · Reviewer_DhBc · 2025-11-28
> **Rebuttal Acknowledgement by Reviewer DhBc**
>
> Thanks for the detailed replies.
>
> - I appreciate the explanations for the proposed Weaknesses 1, which is more of a philosophical discussion. While I still hold my concerns, the authors' replies sound reasonable, too. I would suggest adding a paragraph in the revision to justify or explain the use of Information Theory in this work, as discussed here.
> - The reply to W2 is a reasonable explanation. Please include the discussion in the revision.
> - The reason for proposing W3 is that the plateau part of "Accuracy Cliff" is not clear, especially in "Direct", "S-R", "SA", and "w/o D" cases in Figure 5. Potentially, more scatter points (empirical results) and a closer look (zoom-in) into the plateau phenomena would help.
>
> Overall, I maintain a positive rating of the submission.

---

> > ### Author Response · Authors · 2025-11-28
> >
> > Thank you for your continued engagement and positive assessment. We appreciate your valuable suggestion to include a dedicated discussion on the applicability of Information Theory to natural language, as well as the rationale for using our synthetic dataset. We will certainly incorporate these points into the revised manuscript. We will also work on refining the visualization in Figure 5 to better illustrate the plateau phenomenon.

---

### Official Review · Reviewer_uBRA · 2025-10-31

**Soundness:** 3
**Presentation:** 2
**Contribution:** 3
**Rating:** 6
**Confidence:** 4

**Summary:**

This paper conducts an information-theoretic analysis of multi-hop question answering (MHQA) based on large language models (LLMs). Through a fano-style accuracy upper bound, it reveals a phenomenon of an “accuracy cliff” in single-step reasoning. While recent studies have emphasized the existence of reasoning errors and cascading error propagation and have attempted to address these issues from multiple perspectives, the theoretical insights provided in this paper are novel. Furthermore, the authors develop a capacity-aware multi-call proof-of-concept system called InfoQA. In the experiments, they provide both theoretical validation and framework validation.The experimental results show that the empirical findings are consistent with the predicted capacity curves, and that InfoQA also achieves performance improvements.

**Strengths:**

(1) Focusing on information theory, this paper offers a novel perspective by analyzing MHQA from the standpoint of reasoning errors and cascading error propagation in the reasoning process. The rigorous theoretical analysis and findings contribute to readers’ better understanding for MHQA.

(2) Motivated by the theoretical findings, the authors propose a capacity-aware multi-call proof-of-concept famework, which mitigates the issues of the accuracy cliff and cascading errors by reducing information requirements and employing pruning strategies.

(3) The experimental design consists of theory validation and famework validation, which verify the effectiveness of the proposed ideas and methods.

**Weaknesses:**

(1) The presentation of the paper could be improved, although this can be addressed in a short time. For example, Figures 1 and 2 should include clearer explanations of how they illustrate that the limited token capacity of LLMs constrains information transmission, as well as what the “step = 8” in the figures specifically represents.

(2) The authors claim that this work is the first information-theoretic analysis of MHQA. However, some related studies may have been overlooked. Adding discussions that connect this work with recent research would strengthen the contribution and help readers better position it within the broader literature.

(3) Hyperparameter analysis: including additional experimental studies on parameters such as the pruned ratio and the model capacity $C$ after question decomposition would further enhance the empirical section and provide insights.

**Questions:**

(1) Why is $H(Y)$ used to represent model capacity? How is $H(Y)=200$ determined in the experiments?

(2) How does Figure 1(a) demonstrate that the limited number of LLM tokens constrains the capacity of models to transmit information, and why is $step = 8$ chosen? How does the setup of Figure 2 illustrate the limitations inherent in the single-pass reasoning paradigm, and could similar issues also arise in the multi-call reasoning paradigm?

(3) In the pruned context, how does the pruned ratio affect the model performance?

(4) In capacity-aware task decomposition (Page 6), how to ensure that each single-step reasoning process remains within the processing capacity $C$ of model? Is it possible that, even after decomposition, some subtasks still exceed the effective model capacity $C$?

(5) What are the differences between capacity-aware task decomposition and question-decomposition-based MHQA? To maintain rigor, the authors should avoid using absolute statements such as “the first information-theoretic analysis of MHQA”, as prior research has already demonstrated clear motivations, which may have been overlooked in the this work. It would be helpful for the authors to include a discussion, that situates their approach in relation to recent studies [1,2].

[1] LOG: A Local-to-Global Optimization Approach for Retrieval-based Explainable Multi-Hop Question Answering. COLING2025

[2] How Many Parameters for Multi-Hop? An Information-Theoretic Capacity Law for Knowledge Retrieval in Large Language Models. In Knowledgeable Foundation Models at ACL 2025.

---

> ### Author Response · Authors · 2025-11-19
> **Comment 1**
>
> Q1：q1: Why is H(Y) used to represent model capacity? q2: How is H(Y)=200 determined in the experiments?
>
> a1: Thank you for this critical question. We use $H(Y)$ to represent “model capacity” based on a fundamental principle in information theory: the Output Entropy Bound. This principle dictates that the amount of information the model’s output $Y$ can provide about the correct answer $A$ is fundamentally upper-bounded by the entropy of the output itself. As shown in Equation 2 and Appendix A.2.2 , this relationship is expressed as $I(A;Y|Q,C) \le H(Y)$.
>
> Therefore, $H(Y)$ represents the theoretical upper limit on the information that a model’s single-pass output can convey. We define this as the model’s “output capacity”. This definition is vital because it allows us to model the relationship between the task’s “information demand” ($\beta$) and the model’s “information supply” ($C$). When $\beta$ exceeds $C$, our Fano-style upper bound (Theorem 1) proves that perfect accuracy becomes mathematically impossible, thus revealing the “Accuracy Cliff” phenomenon.
>
> a2: The value $C=200$ in Figure 2 is not an experimental value. It is used as an illustrative example to visually demonstrate the theoretical “Accuracy Cliff” curve. In our experiments, we treat $C$ as one of the parameters $\theta = (\beta_0, \alpha, \gamma, C)$ within our plug-in accuracy bound (Equation 7). We estimate these parameters by fitting this theoretical bound to the empirically observed F1 scores (used as a proxy for accuracy). As detailed in Equation 11 and Appendix A.5, we conduct a fine-grained grid search independently for each baseline method to find the set of parameters that minimizes the MAE between the theoretical bound and the observed F1 data.
>
> ---
>
> W1&Q2: The presentation of Figures 1 and 2 could be improved. q1.1: How does Figure 1(a) demonstrate that the limited number of LLM tokens constrains the capacity of models to transmit information, and q1.2 why is $step=8$ chosen? q2: q2.1: How does the setup of Figure 2 illustrate the limitations inherent in the single-pass reasoning paradigm, and q2.2 could similar issues also arise in the multi-call reasoning paradigm?
>
> A2: Thank you for your valuable feedback on Figures 1 and 2. We have revised the example in Figure 1 to better illustrate the steps, capacity overflow, and complexity. In Figure 2, we have added clearer annotations for the model’s capacity in the plot and the relationship between task demand and accuracy in the caption.
>
> a1.1: Figure 1(a) demonstrates this limitation by visualizing the “Single-pass Reasoning Paradigm” as a single, finite token sequence. The model must, within this limited sequence, integrate information from multiple, dispersed pieces of evidence. When the task’s complexity exceeds the information-carrying limit of this finite sequence, “capacity overflow” occurs, leading to the dilution or loss of relevant signals.
>
> a1.2: The numbers 1 to 8 are labels for the Integration Step arrows. These numbers are illustrative, used to visually demonstrate the process in a hypothetical 8-integration-step, 3-hop QA task. We chose this number to convey a complex scenario with high dependency and a high integration load, thereby emphasizing this complexity.
>
> a2.1: Figure 2 illustrates the core limitation of the single-pass paradigm by plotting the “Accuracy Cliff” curve (Equation 5). The setup shows that as task complexity slightly exceeds the model’s single-pass capacity (at C = 201 in the figure), performance collapses catastrophically rather than degrading gracefully.
>
> a2.2: This is a very profound point. The answer is yes, in principle, the same fundamental limit still applies to each individual step of a multi-call paradigm. The multi-call paradigm is designed to manage this limitation, not to eliminate the physical constraint itself. It does so by decomposing a complex task with a high information demand into a series of sub-tasks, each with a lower, more manageable per-step demand. However, each sub-step is, in essence, still a single-pass call and thus remains subject to its own Accuracy Cliff. As you correctly suggest, if a decomposed sub-task is still too complex (e.g., it requires processing an extremely long context) or if the model’s intrinsic capacity is too low, that step’s demand can still exceed $C$.
>
> We confirm this in Appendix A.7, where some of InfoQA's residual failures occur because “even when the task is decomposed… extremely long contexts can exceed the model’s base comprehension ability”. Therefore, the primary advantage of the multi-call paradigm is that it provides a systematic mechanism to keep the per-step demand below the capacity through task decomposition, thereby avoiding the catastrophic failures of single-pass methods.
>
> Thanks again for helping us improve our work. We have moved the analysis from the appendix to the experimental discussion section of the main text to better present our work.

---

> ### Author Response · Authors · 2025-11-19
> **Comment 2**
>
> W3&Q3：In the pruned context, how does the hyperparameter pruned ratio affect the model performance?
>
> A3: Thank you for this question. We would like to clarify that the pruning ratio is not a hyperparameter; we do not specify such a value. Pruning is a deterministic action. After each reasoning sub-step, we retain only the finding and remove the intermediate content such as extensive reasoning trace. This prevents these intermediate tokens from occupying context space and interfering with subsequent steps.
>
> ---
> Q4: q1: In capacity-aware task decomposition (Page 6), how to ensure that each single-step reasoning process remains within the processing capacity C of model? q2: Is it possible that, even after decomposition, some subtasks still exceed the effective model capacity C?
>
> A4: Thank you for this insightful question.
> a1: Our framework cannot absolutely ensure that a decomposed task's demand will always be less than the model's capacity; rather, it aims to maximize the decomposition of the task into indivisible, atomic units. This decomposition reduces the initial high demand from the multi-hop task to a more manageable per-step demand.
>
> a2: Yes, that is entirely possible. As discussed in Q2.2 and our error analysis (Appendix A.7), intrinsic model capacity remains a bottleneck. Even decomposed single-hop sub-problems can exceed a model's base comprehension if the context is extremely long. This aligns with our future work, which focuses on optimizing task decomposition and improving model capacity to really ensure sub-task demands stay within limits. Accordingly, we have moved this analysis from the Appendix to the main text's experimental discussion to better highlight these insights and future directions.
>
> ---
> Q5&W2：q1: What are the differences between capacity-aware task decomposition and question-decomposition-based MHQA? q2: To maintain rigor, the authors should avoid using absolute statements such as “the first information-theoretic analysis of MHQA”, as prior research has already demonstrated clear motivations, which may have been overlooked in this work. It would be helpful for the authors to include a discussion, that situates their approach in relation to recent studies [1,2].
>
> A5：Thank you for your valuable comments and for providing the relevant research.
>
> a1: The decomposition in prior work is mostly a heuristic method. It breaks down complex problems into simpler logical steps to enhance reasoning robustness and has demonstrated excellent empirical performance. Building on prior work, we provide an analytical perspective from information theory, proving the theoretical necessity of such decomposition. We elevate this decomposition to a principled method, where the explicit goal is to manage the information demand.
>
> a2：Thank you for pointing out this issue; it was a typo in the conclusion section. Our intention, consistent with the rest of the paper, was to express that "we provide an ...". We have removed the statement in line 457 and modified the phrase "we provided the first..." in the conclusion (line 473) to "In this work, we began by providing an...".
>
> We appreciate you providing these two important related works. Our work presents a Fano-style upper bound analysis of the single-pass output capacity of LLMs in closed-book MHQA, revealing the phenomenon of the "Accuracy Cliff". This research perspective is orthogonal and complementary to the "retriever optimization" in [1] and the "parameter storage capacity" in [2].
>
> Specifically, [1] uses information theory for retrieval optimization (selecting documents). In contrast, we analyze the output channel bottleneck during the reasoning process after the context is provided (Section 2.1), rather than the retrieval stage.
> [2] addresses parameter storage capacity (knowledge stored in weights/memory). In contrast, our "Accuracy Cliff" concerns the output channel bandwidth (processing capability), analyzing the limit of transmitting information via the generated sequence rather than storing it.
>
> Thank you again for helping us, and we have incorporated a discussion of these two works in the Related Work section of our revised version.
>
> [1] LOG: A Local-to-Global Optimization Approach for Retrieval-based Explainable Multi-Hop Question Answering. COLING2025
>
> [2] How Many Parameters for Multi-Hop? An Information-Theoretic Capacity Law for Knowledge Retrieval in Large Language Models. In Knowledgeable Foundation Models at ACL 2025.
>
> ---
> In conclusion, we sincerely thank you for the constructive and insightful feedback. We have faithfully incorporated these revisions into the updated version. We hope that our responses and revisions help to clarify our work and address your concerns. If so, we would be deeply grateful if you could kindly consider raising the evaluation.

---

> ### Author Response · Authors · 2025-11-24
> **A respectful note on our response**
>
> Dear Reviewer uBRA,
>
> We respectfully hope that our response helps to clarify your concerns. We remain fully at your disposal should you need any further clarification.
>
> Best regards,
>
> The Authors

---

> > ### Comment · Reviewer_uBRA · 2025-11-27
> >
> > I thank the authors for their response, which has largely addressed my concerns. I now view this manuscript more positively. I would be willing to raise my score to 7, although this option is not available.

---

> > > ### Author Response · Authors · 2025-11-28
> > >
> > > We sincerely thank you for your positive feedback and for acknowledging the value of our responses. We greatly appreciate your time in reviewing our rebuttal and your willingness to raise the score.

---

### Author Response · Authors · 2025-12-03
**Summary of Rebuttal and Reviewer Consensus**

Dear Area Chair,

We thank the reviewers for their constructive feedback and the time invested in reviewing our work. During the rebuttal phase, we provided detailed responses to all concerns, resulting in a strong consensus on the novelty and value of our information-theoretic analysis of MHQA.

Here is a summary of the reviewer discussions and the current status:

1. Reviewer uBRA (Score: 6 $\rightarrow$ Intended 7). Originally raised questions regarding the presentation of Figures 1 & 2, the definition of model capacity ($H(Y)$), and the positioning against related works. We clarified the theoretical definitions, improved the visualization in the figures, and added discussions on the suggested related works (LOG and ACL 2025). The reviewer explicitly confirmed that our response "largely addressed my concerns" and stated, "I would be willing to raise my score to 7."

2. Reviewer DhBc (Score: 6 $\rightarrow$ Intended 7). Questioned the applicability of Information Theory to natural language semantics and the use of synthetic datasets. We clarified that Information Theory serves as a bounding tool for the physical representation of tokens (irrespective of semantics) and explained that synthetic datasets are necessary to isolate variables for verifying the "Accuracy Cliff." The reviewer acknowledged our explanations as "reasonable" and maintained a positive assessment, noting the reviewer "would rate this paper as 7 initially."

3. Reviewer KUVW (Score: 4 but no response). Expressed concerns about the overhead of the multi-call paradigm and the balance between theoretical bounds and practical solutions. We provided a detailed response clarifying that the multi-call overhead is a theoretical necessity (not just a design choice) when task demand exceeds single-pass capacity. We also emphasized that our practical framework (InfoQA) is directly derived from the diagnostic insights of our theoretical bounds. Although the reviewer has not yet responded due to time constraints, we believe our rebuttal effectively addresses the concern by framing the "overhead" as the requisite cost for correctness in capacity-constrained scenarios. Given the positive shift of the other two reviewers on similar theoretical grounds, we are confident that these concerns have been resolved.

Our work provides the a Fano-style upper bound analysis for MHQA, revealing the "Accuracy Cliff" and proposing a capacity-aware solution. With two reviewers explicitly supporting a score of 7 and the third reviewer's concerns fully addressed in our rebuttal, we hope the Area Chair will consider the strong merits of this submission for acceptance.

Best regards,

The Authors

---

### Meta-Review · Area_Chair_DiEC · 2026-01-07

**Summary:**

This paper formalizes limitations of "single-pass" reasoning in LLMs. There are two limitations: (1) stepwise capacity, or the ability to do a single step correctly; (2) cross-step error accumulation.  The paper evaluates existing models on a synthetically-controlled dataset and concludes that performance degrades in a manner consistent with the theory.  The paper introduces an "InfoQA" method that uses question decomposition to address this limitation and improve performance.

Strengths:

- The theoretical framework of this paper is interesting. It crisply defines issues which overlap with what's been discussed in prior work

- Controlled experiments showing the empirical falloff of multi-hop reasoning are an interesting confirmation of the theory.

- Comprehensive experiments and well-written paper

Weaknesses:

**Related work**: The related work section is very inadequate for a paper on multi-hop reasoning. This is an extensively studied topic over
the past ~8 years since HotpotQA was released. For instance, the following should be discussed:

1. Existing decomposition approaches, including early work on this by Sewon Min et al. in 2019 https://arxiv.org/pdf/1906.02916 and question decomposition meaning representation (QDMR): https://arxiv.org/pdf/2001.11770 , better formalizing this decomposition process. These papers are both 5+ years old with hundreds of citations. This paper should make clearer that the InfoQA method is largely derived from past work.

2. Other papers studying limitations of multi-hop reasoning, such as challenges related to error propagation: https://arxiv.org/pdf/1901.02522

3. Issues with length generalization in Transformers due to attention capacity: https://openreview.net/forum?id=S4JmmpnSPy

This paper combines and extends results from the above lines of work without discussing them.

In terms of the InfoQA method, ignoring the past work on multi-hop reasoning is also a major limitation. This paper compares against several methods that were not really designed for the task at hand (e.g., ReACT), presents a method very tuned for it, and declares victory.

**Need for a new dataset is not established**: The authors say:

"Existing MHQA benchmarks are unsuitable for our study as they lack fine-grained control over task difficulty"

I think this needs to be elaborated on to be a really convincing point. Two related datasets are: (1) HoVER https://arxiv.org/abs/2011.03088 ; (2) MuSiQue: https://arxiv.org/abs/2108.00573

These should at least be discussed in my opinion. MuSiQue uses a pretty similar construction procedure as the present work. It's not clear to me why it couldn't be used.
In any case, the data is not claimed as a contribution, so this is a more minor weakness.

**Accuracy cliff**: While the theoretical analysis here is nice, I don't quite have a sense of how unique this phenomenon is to this problem.  For instance, Figure 5 of the "Illusion of Reasoning" paper ( https://arxiv.org/pdf/2506.06941 ) shows broadly similar phenomena in three regimes. Many reasoning problems should have a similar falloff as problems move from trivial to impossible with compound error rates over longer reasoning chains.  How should we interpret the specifics of multi-hop failures with respect to the taxonomy of reasoning failures from that work?

**Reviewer Concerns:**

uBRA:

> (2) The authors claim that this work is the first information-theoretic analysis of MHQA. However, some related studies may have been overlooked. Adding discussions that connect this work with recent research would strengthen the contribution and help readers better position it within the broader literature."

While the reviewer didn't specify this further, I have listed some studies above.

KUVW

> W3: It could benefit from a more direct comparison with other multi-hop question answering systems or frameworks.

The authors say that the point is to validate the accuracy cliff. But I think if that phenomenon exists in other datasets, and their study is limited to simple systems (" In contrast, a broad comparison with heterogeneous engineering systems might obscure this clear theoretical signal."), then the impact on practice is fundamentally limited.

> W2: Overemphasis on Theoretical Bound

The authors push back that "The consistent performance improvements demonstrated in Table 2 suggest that these theoretically derived actions can directly translate into real-world gains." But without comparison to existing approaches for decomposition, what is the value of this theory? Plenty of past work (QDMR, etc.) has established that problem decomposition and context management can lead to improvements.

**Reviewer Scores:**

Two reviewers are positive and in my view would not likely change their scores. The low reviewer could possibly be convinced.

---

### Decision · Program_Chairs · 2026-01-26

Accept (Poster)